# How individual vigor shapes human–human physical interaction

**Dorian Verdel[1]\*, Bastien Berret[2], Etienne Burdet[1]**

[1]Bioengineering Department, Imperial College of Science, Technology and Medicine, London, United Kingdom; [2]Université Paris-Saclay, Inria, CIAMS, Gif-sur-Yvette, France

## eLife Assessment

This is an **important** study showing that movement vigor is not solely an individual property but emerges through interaction when two people are physically linked. The evidence is **convincing**, supported by a well-controlled experimental design and modeling that closely match the observed behavior. While the authors provided a helpful comparison of several candidate models of human-human interaction dynamics, the statistical power remains limited.

**\*For correspondence:**
dverdel@ic.ac.uk

**Competing interest:** The authors declare that no competing interests exist.

**Abstract** The speed of voluntary movements varies systematically, with some individuals moving consistently faster than others across different actions. These variations, conceptualized as vigor, reflect a time–effort–accuracy tradeoff in motor planning. How do two mechanically coupled partners with different individual vigors collaborate, e.g. to move a table together? Here, we show that such dyads coordinate goal-directed movements with minimal interaction force, exhibiting a *dyadic vigor* with similar characteristics as individual vigor. The emerging dyadic motor plan is strongly influenced by the slower partner, whose vigor predicts dyadic vigor, with effects lasting beyond practice. Computational modeling with stochastic optimal control reveals the critical role of partners' movement timing uncertainty and vigor in shaping coordination, allowing us to predict dyadic movements from individual behavior across diverse conditions. These findings shed light on the mechanisms underlying human collaboration and may be used in applications ranging from physical training and rehabilitation to collaborative robotics for manufacturing.

## Introduction

When observing people carrying out actions, one may notice that they move at different paces, with some individuals being systematically slow or systematically fast. Recent research has confirmed this observation (*Labaune et al., 2023*), identifying *vigor*, an idiosyncratic trait across actions that characterizes the tradeoff between the energy expenditure, accuracy, and time to perform a given goal-oriented movement (*Berret and Baud-Bovy, 2022*; *Verdel et al., 2023b*; *Carlisle and Kuo, 2023*; *Thura et al., 2025*), reflecting the value individuals associate with it (*Shadmehr and Ahmed, 2020*). At the neural level, vigor may be influenced by reward-driven dopamine secretion in the basal ganglia (*Nicola, 2010*; *Tachibana and Hikosaka, 2012*; *Wang et al., 2013*; *Rueda-Orozco and Robbe, 2015*; *Jurado-Parras et al., 2020*), where individuals more sensitive to reward tend to move faster. The relative vigor of an individual within a population has been reported to be remarkably stable across repeated measurements compared to its variability between people (*Berret et al., 2018*). Based on this stability, the duration of self-paced human movements can be predicted across tasks and environmental dynamics, as a tradeoff between minimizing a cost of time and the effort required to complete an action with the desired accuracy (*Shadmehr et al., 2010*; *Rigoux and Guigon, 2012*; *Berret and*

*Baud-Bovy, 2022*; *Verdel et al., 2023b*). This idiosyncratic cost of time can be identified using inverse optimal control techniques from the standard relationship observed between motion amplitude and duration (*Berret and Jean, 2016*). In this scheme, the *motor plan*, i.e. the time series of control inputs, emerges from the costs that the central nervous system (CNS) internally minimizes.

When people with different levels of vigor perform a task together, how do they combine their individual motor plans to collaborate efficiently? A pioneering study (*Reed and Peshkin, 2008*) found that *dyads* of connected partners performing point-to-point arm movements tend to move faster than each individual alone. Nevertheless, this result may have been influenced by an explicit reward associated with decreasing movement time given to dyads. Current knowledge suggests that individuals can identify others' vigor through visual observation (*Labaune et al., 2023*), but it is unclear whether this is possible when mechanically connected with a partner through an object, such as when moving a table together. In fact, it has been suggested that during point-to-point movements with a prescribed duration, mechanically coupled individuals may move independently as they do not have sufficiently rich information to identify the partner's motion plan (*Takagi et al., 2016*). Importantly, even if they could coordinate their motor plans when removing the time prescription, the remaining question is whether they would do this in a systematic way, with some dyads moving consistently faster than others across conditions. This paper investigates how dyads of connected partners perform point-to-point movements without time constraints, how partners coordinate their motor plans, and how their individual vigor shapes the joint movement.

Several mechanisms may drive the movement coordination, or absence thereof, between mechanically coupled individuals. First, each individual may execute their motor plan independently, causing large interaction efforts and quite arbitrary movement strategies for the dyad, a scenario we refer to as the *co-activity* hypothesis (*Takagi et al., 2016*). However, one could also expect that, when time is not prescribed, a leader naturally emerges and imposes their preferred motor plan to the dyad. Specifically, a commonly suggested interaction strategy (*Losey et al., 2018*) is that the slower (respectively, faster) individual in a dyad could infer and align with the faster (slower) partner's movement strategy (*Reed and Peshkin, 2008*; *Groten et al., 2009*; *Melendez-Calderon et al., 2015*), which we refer to as *leader–follower* hypothesis. More recent studies have shown that connected individuals can use interaction forces to exchange their motion plans and both improve performance in a tracking task (*Takagi et al., 2017*; *Takagi et al., 2019*). These results suggest that connected individuals may co-adapt to establish a common motor plan using haptic communication. A negotiated motor plan could then emerge, where both partners contribute to setting the dyad's behavior, based on a weighted average of their nominal vigor in the task. Hence, both partners may similarly contribute to the dyad's strategy by blending their original intents, a hypothesis we refer to as *weighted adaptation*. Alternatively, uncertainty about a partner's motor plan could change an individual's own motor plan and vigor, leading to an *interactive adaptation* hypothesis. According to this view, the dyad's motor plan would not originate from a weighted averaging of the partners' initial strategies, but from a dynamic change of their strategies in response to the current interaction dynamics, which can be represented in cost functions. These four hypotheses, schematized in *Figure 1A*, yield distinct behavioral predictions that will be tested in the present study.

## Results

To understand how dyads' partners coordinated their motor plans and test the above hypotheses, we investigated point-to-point movements without timing constraint or explicit reward. Relatively large targets were used in order to relax accuracy constraints. The partners carried out these movements while being connected through a virtual elastic band implemented using individual wrist exoskeletons, as illustrated in *Figure 1B*. This setup allowed us to systematically investigate the impact of the interaction on movement planning, by varying movement amplitudes and connections' stiffness, as well as the effort associated with movements using a viscous load. Twenty participants (10 dyads) were requested to perform self-paced flexion and extension movements of the right wrist, with amplitudes between 18° and 90°, by controlling a cursor on their individual monitors from $A_0$ toward one of the targets $\{A_1, A_2, A_3, A_4, A_5\}$ and back to $A_0$. The individual cursor position was an affine function of the corresponding wrist angle. The two partners of a dyad first performed movements in a *solo session* so that we could identify their individual motor strategies, in particular their vigor, prior to being connected in a *dyadic session* with an elastic band of stiffness $\kappa = 0.5\,\mathrm{Nm/rad}$ (KL) or $\kappa = 1.6\,\mathrm{Nm/rad}$

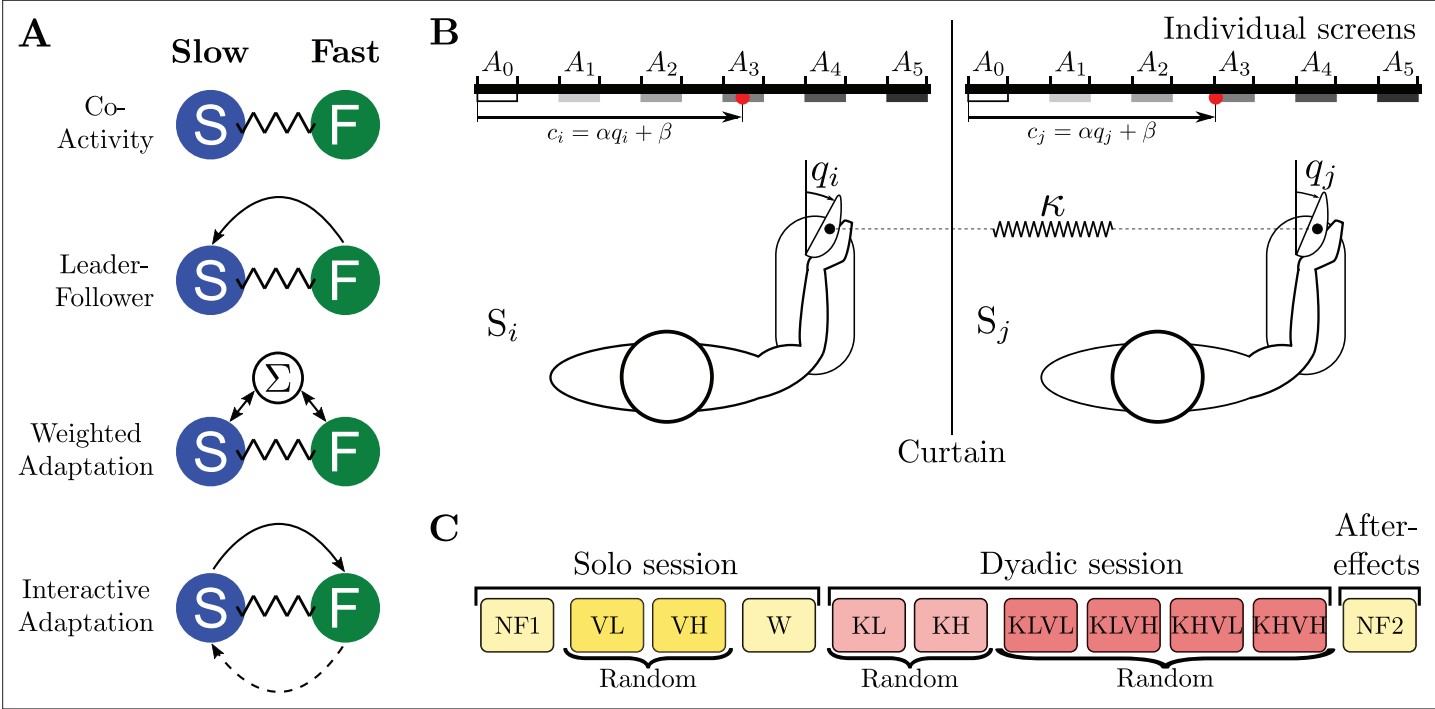

**Figure 1.** Hypothetical strategies to combine motor plans and experiment. (**A**) Different hypotheses for coordination between a fast (F) and a slow (S) partner. The arrows represent the haptic communication flow triggering motor adaptations in the respective hypotheses. The *co-activity* strategy corresponding to independent motor plans, *leader–follower* here based on the faster partner's individual motor plan, *weighted adaptation* generalizing the leader-follower hypothesis based on a weighting of their initial strategies in the task, and *interactive adaptation* where both partners dynamically adapt their original motor plan, possibly differently as illustrated by the solid and dashed arrows, due to the interaction with an uncertain partner. (**B**) Experiment to investigate vigor in individuals and in mechanically connected dyads. The two partners have to reach one of the targets $\{A_1, ..., A_5\}$ on their individual monitor using wrist flexion/extension of the right arm. Their real-time wrist angles $q_i$ and $q_j$ are mapped to individual red cursors $c_i$ and $c_j$. In the dyadic session, their hands are coupled through a virtual elastic band of either low (KL) or high (KH) stiffness. (**C**) Experimental protocol. The initial solo session consists of four blocks: one null-field block (NF) with exoskeletons' motor off to estimate the individual vigor, two blocks with low (VL) or high (VH) resistive viscous load to vary the cost of movement (i.e. effort), and a null-field washout block (W). The subsequent dyadic session involves six blocks in coupled mode and a final block in null-field mode to analyze after-effects of the practice with mechanical connection. The KL and KH blocks, performed in random order, first allow participants to familiarize themselves with the interaction, and the KxVx blocks are to investigate all combinations of the connection stiffness and viscous load.

(KH). To facilitate reading, the figures show results obtained with the low-stiffness connection KL (except when the two stiffness and viscous loads led to different behaviors), while complementary figures with large connection stiffness KH and other parameters referred to in the text are provided in the supplementary materials.

## Solo session

We first analyzed participants' behavior in the solo session to verify that vigor, as defined in previous studies (*Berret et al., 2018*; *Labaune et al., 2020*; *Berret and Baud-Bovy, 2022*), is an idiosyncratic trait within our sample, and to identify their motor plan as a baseline for comparison and modeling in the dyadic session. As illustrated in *Figure 2A*, the average trajectories in the *null-field condition* (NF1, with motor off) exhibit a bell-shaped velocity profile, where movement duration increases with amplitude. Submovements near the target appear for the farther targets $\{A_4, A_5\}$, as was also observed in self-paced movements of large amplitude carried out with another exoskeleton (*Verdel et al., 2023b*).

Since vigor is expected to be a stable individual trait across effort levels (*Berret et al., 2018*; *Labaune et al., 2020*), we first verified that VL and VH imposed sufficient resistance to influence movement duration (Friedman: $W = 0.65$, $p < 10^{-3}$, *Figure 2—figure supplement 1A*). Participants moved significantly slower in VL and VH compared to NF1, and slower in VH than in VL ($p < 2 \cdot 10^{-3}$; Cohen's $D > 0.56$ for all of VL–NF1, VH–NF1, and VL–VH). These findings confirm that the selected

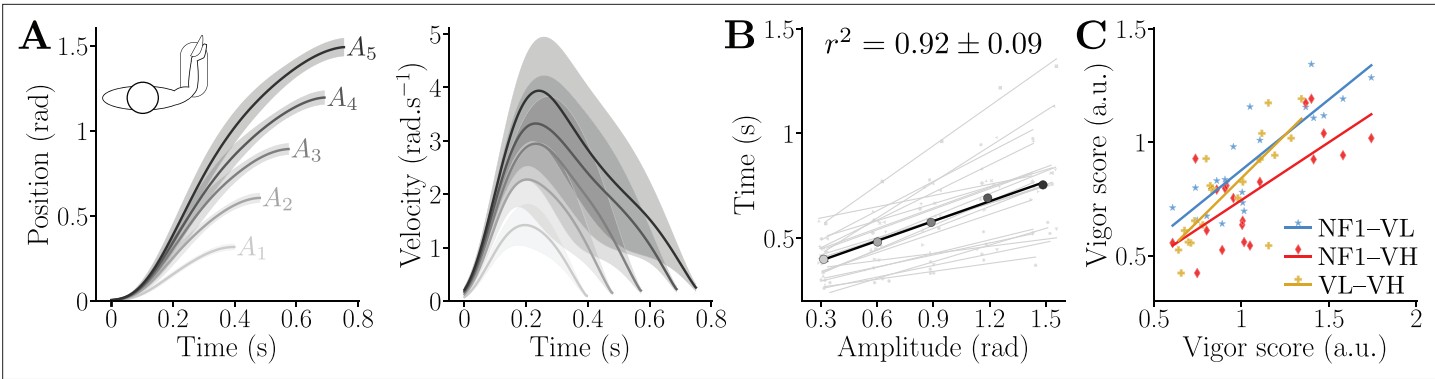

**Figure 2.** Kinematics and vigor in the solo conditions. (**A**) Trajectories of reaching movements of different amplitude averaged across the population in the first null-field block (NF1) of *Figure 1C* (shaded areas represent the standard deviations). (**B**) Resulting individual and averaged amplitude–duration relationships. (**C**) The vigor scores in the three solo conditions {NF1, VL, VH} are all correlated, indicating that vigor can be defined robustly across varied effort conditions.

The online version of this article includes the following figure supplement(s) for figure 2:

**Figure supplement 1.** Effect of the viscous resistance on movement time and vigor.

**Figure supplement 2.** Minimum time–effort compromise in the VL and VH conditions.

viscous loads required sufficient effort to impact movement duration and analyze the robustness of inter-individual differences.

Next, as shown in *Figure 2B*, movement time increased linearly with amplitude for each participant ($r^2 = 0.92 \pm 0.09$) and for the average participant in our sample ($r^2 = 0.99$). We leveraged the extracted affine amplitude–duration relationships to compute a vigor score $v_i$ for each participant relative to the population (*Choi et al., 2014*; *Labaune et al., 2020*; *Verdel et al., 2023b*):

$$v_i = \frac{\sum_{k=1}^{5} \overline{T}^2(A_k)}{\sum_{k=1}^{5} T_i(A_k)\,\overline{T}(A_k)} \,, \tag{1}$$

where $\overline{T}(A_k)$ is the average movement duration of the population for the *kth* movement amplitude in the relevant null-field condition (NF1 for the solo session, KL or KH for the dyadic session), and $T_i(A_k)$ is the average movement duration of participant $i$ with amplitude $A_k$ in the considered condition. The vigor score is positive, and a participant moving at the same speed as the population average has a unit vigor.

A decomposition of variance revealed that inter-individual variability of the vigor scores through the NF1, VL, and VH conditions accounted for 81.2% of the total variance, despite the possible effects of the viscous load on the total variance. This result is consistent with prior experiments testing the timing stability over multiple sessions (*Berret et al., 2018*), rather than by varying the effort in the present study. The vigor scores for NF1, VL, and VH exhibit distributions similar to previous studies (*Labaune et al., 2020*; *Verdel et al., 2023b*; *Figure 2—figure supplement 1B*) and correlate across conditions (*Figure 2C*; $p < 10^{-3}$; $r > 0.71$, Pearson tests), confirming the robustness of the individual vigor score across effort levels.

In summary, the solo session data – characterized by affine amplitude–duration relationships, greater inter- than intra-individual variability, and stable vigor scores across conditions with varied effort levels – confirm that vigor is an idiosyncratic trait in our sample of participants, consistent with prior findings (*Berret et al., 2018*; *Labaune et al., 2020*).

## Dyadic session
### Coordination of motor plans within dyads
To analyze coordination, we identified the slow and fast partner in each dyad based on their vigor in the first null-field block, thereby forming the *slow partners* group and *fast partners* group. We first

examined whether reaction times differed between these two groups in the initial null-field block, as it could naturally lead to a leader–follower behavior, with the leader being the participant starting earlier. However, no significant difference was observed ($p > 0.14$, reaction time about $225\,\text{ms}$ on average), thus the reaction time did not depend on the group in our task.

We then assessed whether mechanical coupling affected the participants' reaction time in KL and KH. We found a main effect of the condition on the reaction time for both groups (in both cases: $W > 0.39$, $p < 0.02$). In KL, reaction times were not significantly impacted. In KH, the reaction times of the slow group were significantly shorter than in the initial null-field block ($p = 0.03$, $D = 1.46$), with a similar trend observed for the fast group ($p = 0.06$, $D = 1.03$). However, the fast and slow participants still had similar reaction times ($p > 0.34$ for both KL and KH), with an average of $190\,\text{ms}$.

Regarding trial-by-trial adaptations, the only clear observed trend was an increase in the interaction force during the initial movements, most pronounced in KH and for the largest amplitude in KL (see *Figure 3—figure supplement 1*). This increase in force was not accompanied by changes in movement duration, although slight decreasing trends were occasionally visible across trials (see *Figure 3—figure supplement 2*). These results suggest that participants maintained a minimal level of interaction force to facilitate synchronization.

To test the *co-activity* hypothesis, we modeled the fast and slow partners as planning trajectories independently with their respective preferred movement duration. The individual *co-activity* trajectories and interaction force were computed by assuming participants optimally tracked their desired trajectory with minimum effort and error, while connected via an elastic band of stiffness KL or KH (see Methods) (*Todorov and Jordan, 2002*). The averaged partners' trajectory (solid black line in *Figure 3A*) exhibits a non-smooth slope change, due to the submovement and stabilization of the fast partner's within the target (solid green curve in *Figure 3A*), and produced a relatively large interaction torque that increased with vigor difference between the partners.

The data of connected participants in our experiment did not correspond to these predictions. First, averaged participants' trajectories in the connected session exhibited bell-shaped velocity profiles smoother than in solo performances (*Figure 3B*), as analyzed with spectral smoothness metrics (*Balasubramanian et al., 2015*; see *Figure 3—figure supplement 3*). Second, the absolute interaction torque was low across all conditions, irrespective of target, viscosity, or connection stiffness (*Figure 3C*, *Figure 3—figure supplements 4C and 5–14*). It was slightly increasing with movement amplitude with the low connection stiffness (significant effect of the condition and interaction with the target: $W > 0.32$, $p < 0.013$, confirmed by pairwise comparisons, $p < 0.02$, Cohen's $D > 0.36$ when pooling all KL conditions for each target) but was always less than $0.1\,\text{Nm}$ ($p < 0.012$, Cohen's $D > 3.73$ in the six connected conditions), and at maximum close to $0.05\,\text{Nm}$ for target $A_5$ with KL. Interestingly, interaction torques were higher for KL than KH (in all cases: $p < 0.002$, $D > 1.34$), likely due to deteriorated haptic communication with a more compliant connection (*Takagi et al., 2018*). Furthermore, interaction torque showed no significant correlation with differences in individual vigor within dyads (KL conditions: $|r| < 0.43$, $p > 0.22$, KH conditions: $|r| < 0.18$, $p > 0.61$; *Figure 3D*; *Figure 3—figure supplement 4D*). In contrast to these results, the *co-activity* hypothesis predicted average interaction torques for $A_5$ of $0.1\,\text{Nm}$ with the KL connection (i.e. around twice higher than observed) and $0.23\,\text{Nm}$ with KH (i.e. around ten times higher than observed). These findings rule out the *co-activity* hypothesis. They rather suggest that dyadic partners closely coordinated their motor commands, and that at least one of them adapted their individual motor plan.

## Characteristics of individual movements within connected dyads

As shown in *Figure 3E* (and *Figure 3—figure supplement 4B*), the movement duration of connected participants increased linearly with amplitude, both with low and high connection stiffness, for each participant (KL: $r^2 = 0.85 \pm 0.21$; KH: $r^2 = 0.92 \pm 0.09$) as well as for the average participant (KL: $r^2 = 0.98$; KH: $r^2 = 0.99$). Furthermore, the individual movement duration of connected participants was affected by VL and VH loads ($W > 0.85$, $p < 10^{-3}$ for both KL and KH conditions, as illustrated by *Figure 3F*; *Figure 3—figure supplement 4E*). As in solo trials, dyads moved slower when $\nu > 0$ ($p < 10^{-3}$, Cohen's $D > 1.72$ in all cases) and with VH compared to VL ($p < 10^{-3}$, Cohen's $D > 1.02$ for both KL and KH). Despite potential difficulties in coordinating with the partner, movement duration was not affected by connection stiffness, independently of the required level of effort. Moreover, movement duration was not significantly different between dyads' partners (*Figure 3G*; *Figure 3—figure*

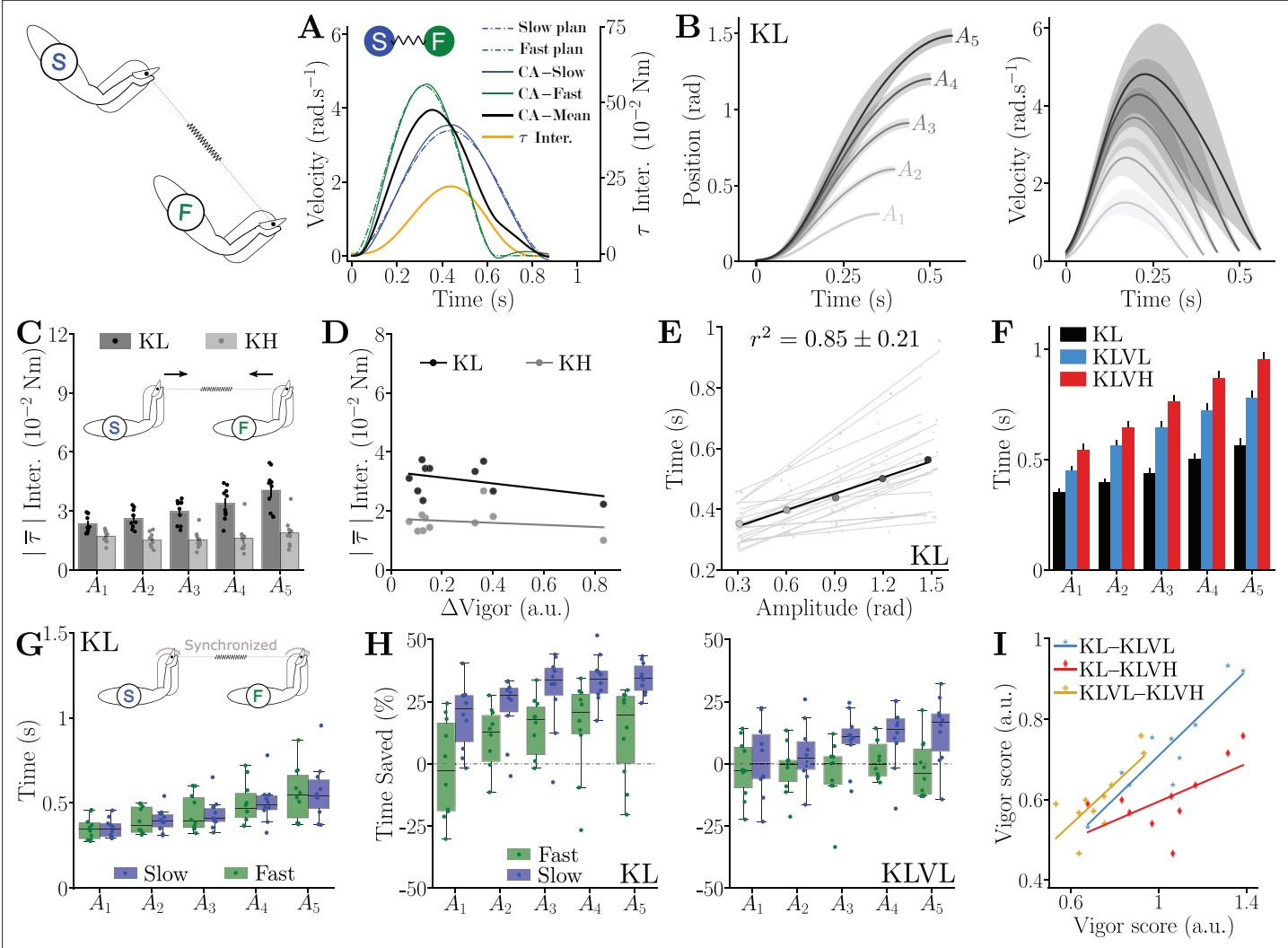

**Figure 3.** Dyads' kinematics, interaction torque, and dyadic vigor. (**A**) Predicted trajectories and interaction torque of a dyad with independent motor plans for the faster and slower partners for $A_5$ in the KL condition. (**B**) Participant trajectories averaged across the population for KL and for the five targets (shaded areas represent the standard deviations). (**C**) Average absolute interaction torque with low (KL) and high (KH) connection stiffness. Error bars represent standard errors. (**D**) The average interaction efforts in KL and KH are independent of the difference in individual vigor between the partners of a dyad (here during the first null-field block). (**E**) Amplitude–duration relationships of each participant's movements and average relationship across the population during the KL condition. (**F**) Effect of the two different viscous loads on movement duration, for the KL connection, averaged across all participants with error bars representing standard error. (**G**) Movement durations are not different between the fast and slow groups in the connected conditions (here KL). (**H**) Percentage of time saved by the fast and slow groups between NF1 and KL, and between VL and KLVL, where positive values indicate faster movements. (**I**) Correlations between the vigor scores obtained during the three KL connected conditions. Vigor scores were computed using the average movement duration between members of the dyad.

The online version of this article includes the following figure supplement(s) for figure 3:

**Figure supplement 1.** Adaptation of interaction torque through trials in the connected session.

**Figure supplement 2.** Limited adaptation of movement duration through trials in the connected session.

**Figure supplement 3.** Movement performance of the dyads.

**Figure supplement 4.** Complementary behavioral data of the dyadic session.

**Figure supplement 5.** Average interaction efforts throughout conditions and targets for dyad D1.

**Figure supplement 6.** Average interaction efforts throughout conditions and targets for dyad D2.

**Figure supplement 7.** Average interaction efforts throughout conditions and targets for dyad D3.

**Figure supplement 8.** Average interaction efforts throughout conditions and targets for dyad D4.

**Figure supplement 9.** Average interaction efforts throughout conditions and targets for dyad D5.

*Figure 3 continued*

**Figure supplement 10.** Average interaction efforts throughout conditions and targets for dyad D6.

**Figure supplement 11.** Average interaction efforts throughout conditions and targets for dyad D7.

**Figure supplement 12.** Average interaction efforts throughout conditions and targets for dyad D8.

**Figure supplement 13.** Average interaction efforts throughout conditions and targets for dyad D9.

**Figure supplement 14.** Average interaction efforts throughout conditions and targets for dyad D10.

**Figure supplement 15.** The fast and slow participants have similar movement durations when connected.

---

*supplement 15*, no main effect, $p = 0.86$). Altogether, these results indicate that the viscous loads were sufficient to impact the dyads' behavior, and dyads were able to plan and coordinate their movements efficiently in all conditions.

A striking observation is that, when using KL and KH without viscous load, connected participants from both the slow and fast groups moved faster than in the solo session ($W = 0.49$, $p < 10^{-3}$), with a greater change in movement duration for the slow partners (Kruskal $H = 106$, $p < 10^{-3}$ see *Figure 3H*; *Figure 3—figure supplement 4F*). Post hoc comparisons confirmed this effect ($p < 10^{-3}$, Cohen's $D > 0.79$ for both groups). With the resistive load, the slow partners still saved time compared to solo trials (*Figure 3H*; *Figure 3—figure supplement 4G–I*, $W = 0.55$, $p < 10^{-3}$), whereas the fast partners' movement durations were not significantly different from the solo conditions (*Figure 3—figure supplement 4H, I*). Post hoc comparisons confirmed that the slow partners saved time for all the connected conditions ($p < 0.008$, Cohen's $D > 0.71$ in all cases). Importantly, regardless of whether dyads saved time or not, it did not affect their movement smoothness or accuracy (see *Figure 3—figure supplement 3* and associated analyses in Appendix 1).

In summary, dyads moved faster than lone individuals when there was no viscous load. This increased movement speed was a strong marker of unloaded dyadic movements, with, for instance, the slow partners moving 30% faster for movements to target $A_5$ and the fast partners 20% faster (medians in *Figure 3H*). With a viscous load, the slow group saved time while the fast group kept their movement duration constant (medians in *Figure 3H*). The combination of all these results on interaction efforts and time supports the existence of a shared motor plan in dyad members, which plan is sometimes clearly differing from the lone motor plan of both participants.

## Dyadic vigor

As connected partners coordinated their movements and exhibited affine amplitude–duration relationships, we computed vigor scores based on the average movement duration of the two partners of each dyad using *Equation 1*. As for the solo session, we analyzed the proportion of variance in vigor explained by inter-dyad differences throughout the connected session. When grouping all the connected conditions, we found that intra-dyad (that is, variability of vigor of a dyad across blocks) variability explained 63% of the total variance. However, when analyzing the blocks with and without viscous load separately, inter-dyad variability explained 99.4% (KL and KH) and 72.3% (in the four loaded conditions) of the variance. This means that the viscous loads changed how vigor scores are spread, thereby increasing intra-individual variability. However, when comparing conditions with similar dynamics, the variability in dyadic vigor scores was primarily accounted for by differences between dyads, as for lone movements. Therefore, each dyad seemed to behave as a single unit with its own preferred vigor.

To confirm this analysis, we then examined whether the relative dyadic vigor scores across the dyads changed with load and connection stiffness using Pearson correlation coefficients between pairs of conditions. All 15 Pearson correlations were significant ($p < 4.6 \cdot 10^{-3}$, Pearson-$r > 0.64$ in all cases), showing that the order of dyads according to their vigor scores was robust to evolving dynamics and effort levels (see *Figure 3I*; *Figure 3—figure supplement 4J* for examples of correlations). This confirms that vigor scores mainly vary across dyads rather than between conditions.

Therefore, we conclude that goal-oriented movements of mechanically connected individuals are characterized by a *dyadic vigor*, supported by (1) coordinated motor plans between partners, evidenced by minimal interaction effort and movement duration differences; (2) kinematics similar to individual movements, with overall higher performance (time, smoothness, and accuracy); (3) affine

amplitude–duration relationships; (4) larger inter-dyad than intra-dyad variability; and (5) vigor scores robustly correlated throughout all conditions.

## Emergence of dyadic vigor from the partners' individual vigor

Connected movements durations were either shorter than those of the fast partners, in the unloaded conditions, or close to them in the loaded conditions (*Figure 3H*, *Figure 3—figure supplement 4G–I*). This suggests that the faster partner primarily determined the dyadic vigor, effectively acting as a leader. Such behavior rules out the *leader–follower* hypothesis where the slower partner would lead. Conversely, the *leader–follower* hypothesis assuming the fast partner as leader, where the slow individual merely aligns their motor plan based on interaction forces, appears plausible but fails to explain the observed increase of movement velocity in unloaded conditions.

An alternative interpretation is that the fast partner indeed leads but also adjusts for the slower partner's vigor when setting the dyadic vigor, for instance via a convex combination with a higher weight on their own vigor. According to this *weighted adaptation* hypothesis, the dyadic vigor $v^d$ would depend on the vigors $v^f$ and $v^s$ of the fast and slow partners as:

$$v^d = \alpha v^f + (1 - \alpha)v^s, \tag{2}$$

where $\alpha$ allows to adjust the importance given to each partner: $\alpha = 1$ and $\alpha = 0$ fall back to the *leader–follower* hypothesis with a fast and a slow leader respectively, while $\alpha = 0.5$ represents equal contribution. Values $\alpha \in (0.5, 1)$ indicate a dominant fast leader marginally accounting for the slow partner's vigor, and conversely for $\alpha \in (0, 0.5)$. Based on the movement duration results, $\alpha$ would likely be close to 1 in our dataset.

To evaluate this hypothesis, we fitted a linear mixed model (LMM) assessing how dyadic vigor depends on the vigor of the fast and slow partners across conditions (see *Equation 5* in Methods). Surprisingly, dyadic vigor was significantly related to the fast partners' vigor only in KLVL ($r = 0.14$, $p = 0.03$) and not in any other condition (KL and KH: $r < 0.14$, $p > 0.22$; KHVL: $r = 0.13$, $p = 0.15$; KLVH and KHVH: $r < 0.1$, $p > 0.24$). In contrast, the slow partners' vigor significantly predicted dyadic vigor in five out of six conditions (KL and KH: $r > 0.624$, $p < 0.001$; KLVL and KHVL: $r > 0.24$, $p < 0.009$; KLVH: $r = 0.2$, $p = 0.028$).

This unexpected result invalidates the *leader–follower* hypothesis, which predicts that dyadic vigor should primarily depend on the fast partner's vigor. Moreover, the LMM analysis also indicates that dyadic vigor is not influenced by mixed parameters, such as the difference in vigor between partners or a convex combination of their individual vigors as in *Equation 2*. In that case, the fast partner's vigor would have been a significant predictor. In conclusion, this refutes the *weighted adaptation* hypothesis, which assumes similar contributions from both partners to the dyadic vigor.

We are thus left with the *interactive adaptation* hypothesis, according to which both participants adapt their control strategy to the new dynamical context induced by the interaction with the partner. While individuals may not perfectly infer their partner's motor plan, they may identify the distribution of the partner's movement durations and plan their motion accordingly. In particular, each participant may partially anticipate the partner's timing and adjust their own vigor to minimize interaction efforts variability, which otherwise leads to instability. The effect of uncertainty induced by the partner's timing, accumulating over time, may prompt individuals to move faster than initially planned – consistent with evidence that higher speeds mitigate the detrimental effects of uncertainty on performance (*Berret et al., 2021*). This dynamic change of dyadic vigor could explain why dyads move even faster than the faster partner in unloaded conditions.

## Computational modeling of coordination mechanism

To test this premise, we designed a computational model representing this dynamic adaptation related to the partner's uncertainty. Based on the above observations, it aims at predicting (1) minimal variation of interaction efforts, as dyads were shown to coordinate their movements to allow the emergence of dyadic vigor, (2) dyads moving faster than either partners in solo (when there is no load), (3) dyads' movement times close to those of the fast partner in solo in presence of load, (4) a dyadic vigor that depends mainly on the vigor of the slow partner, and (5) small dyadic movement duration prediction errors across the 30 tested conditions (5 targets × 2 stiffness levels × 3 viscous loads).

We simulated a dyad that adapts its behavior relative to an inferred distribution of the slow partner's movements (see Methods for details), while integrating the costs associated with the fast partner during planning. Accordingly, the model assumes asymmetric roles for the two partners, where the fast partner costs are used by the dyad to plan the movements while the uncertainty arising from the interaction with the slow partner acts as a higher level constraint through the interaction torque. Conceptually, this asymmetry reflects the role of the slow partner in setting the vigor ranking within the dyads, and the role of the fast partner in determining the average dyadic behavior.

First, we extract an optimal control model of effort and time characterizing the participants' average behavior from data of the solo session (as explained in the Methods) (*Berret and Jean, 2016*). Using the identified cost function, we can predict the movement duration and mechanical work of the average lone participant in the VL and VH conditions based on their initial null-field behavior (see *Figure 2—figure supplement 2*). This identified individual motor plan serves as a basis for simulating the interaction within dyads. Specifically, the effort cost identified for an average subject is used to build the model associated with the *interactive adaptation*, while the time cost is identified separately for the slow and fast groups (see Methods for values of fitted parameters). To model the interaction uncertainty, the slow partner is represented by a distribution of minimum jerk trajectories of stochastic duration $T_k^s = \xi_k$ with average $\mu_k$ and deviation $\sigma_k$ for target $k$, based on their average vigor. The dyads' common plan is then computed through the minimization of the expectation of a cost function considering the common effort cost and the individual cost of time of the fast partner, as well as the variations of interaction torque $\dot{\tau}_k^i$ :

$$C_k(u_k, T_k) = \mathbb{E}\left[ V_k^f(u_k, T_k) + Q_\tau \int_0^{T_k} g(\dot{\tau}_k^i; \xi_k)\, dt \right] ,$$  (3)

where $\mathbb{E}$ is the expectation with respect to the slow partner's distribution, $V_k^f$ the cost function including the common cost of effort and the cost of time of the fast participant (see *Equation 10* in Methods), $Q_\tau$ a weight and $T_k$ the free final time minimizing a positive function $g(\dot{\tau}_k^i; \xi_k)$ of the interaction torque variation $\dot{\tau}_k^i$, associated with the distribution of the slow partner's movements duration. The computation and optimization of the cost function are described in the Methods.

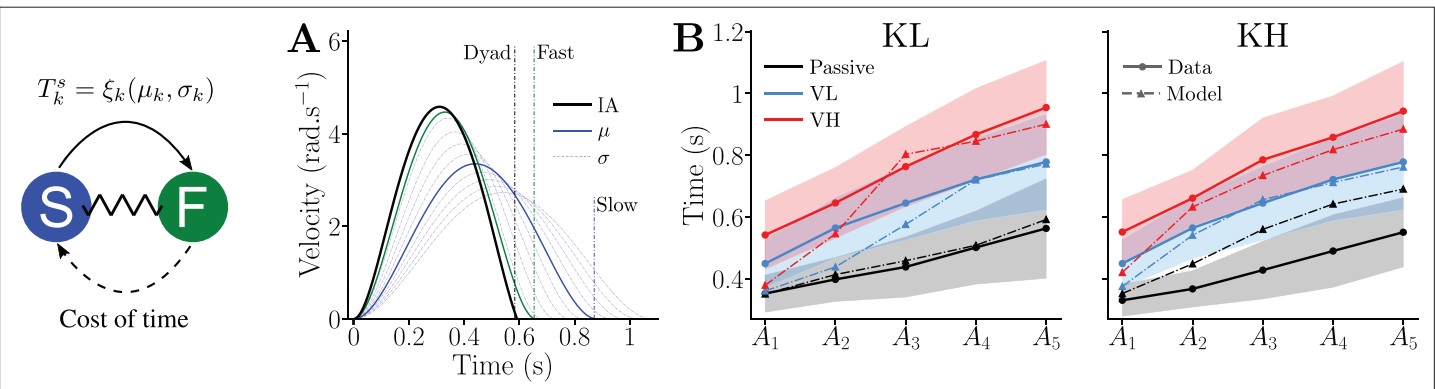

**Figure 4.** Predictions of dyad movement duration obtained with the model arising from the *interactive adaptation* hypothesis. (**A**) Illustration of predicted velocity profiles and measured averaged durations for movements toward $A_5$ in the KL condition. The solid and dashed-dotted blue lines represent, respectively, the average and samples of the distribution of the slow partner, with the average movement duration of the slow partners as a dashed-dotted blue vertical line. The solid green line illustrates the average velocity profile of the fast partners moving alone, with the green dashed-dotted vertical line the corresponding duration in the experiment. Finally, the solid black represents the velocity profile predicted by an *interactive adaptation* (IA), and the black dashed-dotted vertical line the average experimental duration of dyadic movements. (**B**) Comparison between predicted movement time and data for all the connected conditions.

The online version of this article includes the following figure supplement(s) for figure 4:

**Figure supplement 1.** Evolution of optimal time predictions for the KL conditions.

**Figure supplement 2.** Evolution of optimal time predictions for the KH conditions.

**Figure supplement 3.** Effect of varying the cost of time of the fast partner on predicted movement time.

**Table 1.** Ability of the different models to reproduce the main qualitative (top) and quantitative (bottom) experimental findings. The interactive adaptation hypothesis is the only one among those tested that can reproduce all the main qualitative experimental findings. It also results in the lower median prediction errors for the low stiffness conditions and in the second lower median errors for the high stiffness movements, with a negligible $12\,\mathrm{ms}$ difference compared to the best predictions.

| Characteristic | Co-activity | Leader–follower Fast lead | Leader–follower Slow lead | Weighted average | Interactive adaptation |
|---|---|---|---|---|---|
| Bell-shaped and smooth velocity profile | No | Yes | Yes | Yes | Yes |
| Low h–h interaction force | No | Yes | Yes | Yes | Yes |
| Dyad faster than fast partner | No | No | No | No | Yes |
| Only slow partner impacts h–h vigor | Yes | No | Yes | No | Yes |
| Median error low stiffness ($\mathrm{ms}$) | 138.2 | 35 | 138.2 | (35, 138.2) | 28.7 |
| Median error high stiffness ($\mathrm{ms}$) | 130.5 | 38 | 130.5 | (38, 130.5) | 50.8 |

The model's predictions, summarized in **Figure 4**, closely match the experimental data in both the bell-shaped, smooth velocity profile and the predicted movement durations. The median prediction errors across the 30 simulated conditions were small (KL: $28.7\,\mathrm{ms}$; KH: $50.8\,\mathrm{ms}$) with most errors below $80\,\mathrm{ms}$ except for $A_1$ with viscosity, $A_2$ with KL and viscosity, and $A_3$–$A_5$ for KH without viscosity. Importantly, all these predictions were obtained using the same weighting $Q_\tau$ in the cost function in **Equation 3** (or **Equation 17** in the Methods for details). To summarize the evaluation of plausible hypotheses for the mechanism of coordination within a dyad, **Table 1** presents a systematic comparison of the four tested hypotheses on their ability to reproduce both the qualitative and quantitative experimental findings. This analysis further supports the validity and relevance of the *interactive adaptation* hypothesis, and its implementation as an asymmetric model where the slow partner acts as an overarching constraint while the fast partner acts on the expected dyad behavior, compared with the alternative explanations tested.

How does the *interactive adaptation* model work? Co-adaptation of the two partners yields an average dyadic movement duration close to the average preferred duration of the fast group, but the slow partner ultimately determines which dyad is faster or slower relative to this average behavior. In our approach, the uncertain term associated with the slow partners should be understood as an overarching constraint that conditions the strategy of the dyad, while the fast partner cost of time acts as a contributor to the expected dyad strategy. Conceptually and numerically, as reported in the sensitivity analysis, this asymmetry corresponds to the role of the slow partners in setting the vigor ranking among the dyads and the role of the fast partner in setting the average dyadic behavior. Importantly, the uncertainty associated with the distribution of the slower partner's motor plan may explain the faster movement in KL and KH. Specifically, the partners would try to minimize the accumulation of errors in expected variations of interaction efforts to stabilize the interaction, resulting in faster movements as this cost naturally increases with time (see uncertainty term in **Equation 17** in Methods). In this context, the slower partner would only be willing to incorporate this acceleration up to a certain point depending on their individual vigor, thereby determining the final dyadic vigor. This would also explain why this global acceleration is not observed in the presence of a viscous load. Although such efforts may increase the deviation of the inferred partner's distribution by masking interaction efforts, they also naturally increase stability by dissipating energy. Consequently, they automatically reduce the variations in interaction efforts and their associated detrimental effects on dyadic dynamics, thereby reducing the cost of increased movement duration. In our model, this important property naturally emerges from the compromise between time and effort, as identified for individual behavior, and uncertainty, due to the interaction with the partner, when controlling the dyad's dynamics.

**Figure 4—figure supplements 1 and 2** present the results of a sensitivity analysis varying the model's three parameters: weight of interaction torque variation minimization and the slow partner's movement time distribution average and deviation $T_k^s = \xi_k(\mu_k, \sigma_k)$. This analysis shows that (1) the model remains stable beyond the durations and parameters examined in the present study, and (2) dyadic movement duration decreases as $\mu_k$ decreases for large ranges of parameters, consistent with the observed correlation between dyadic vigor and the slow participant's vigor. We also verified the

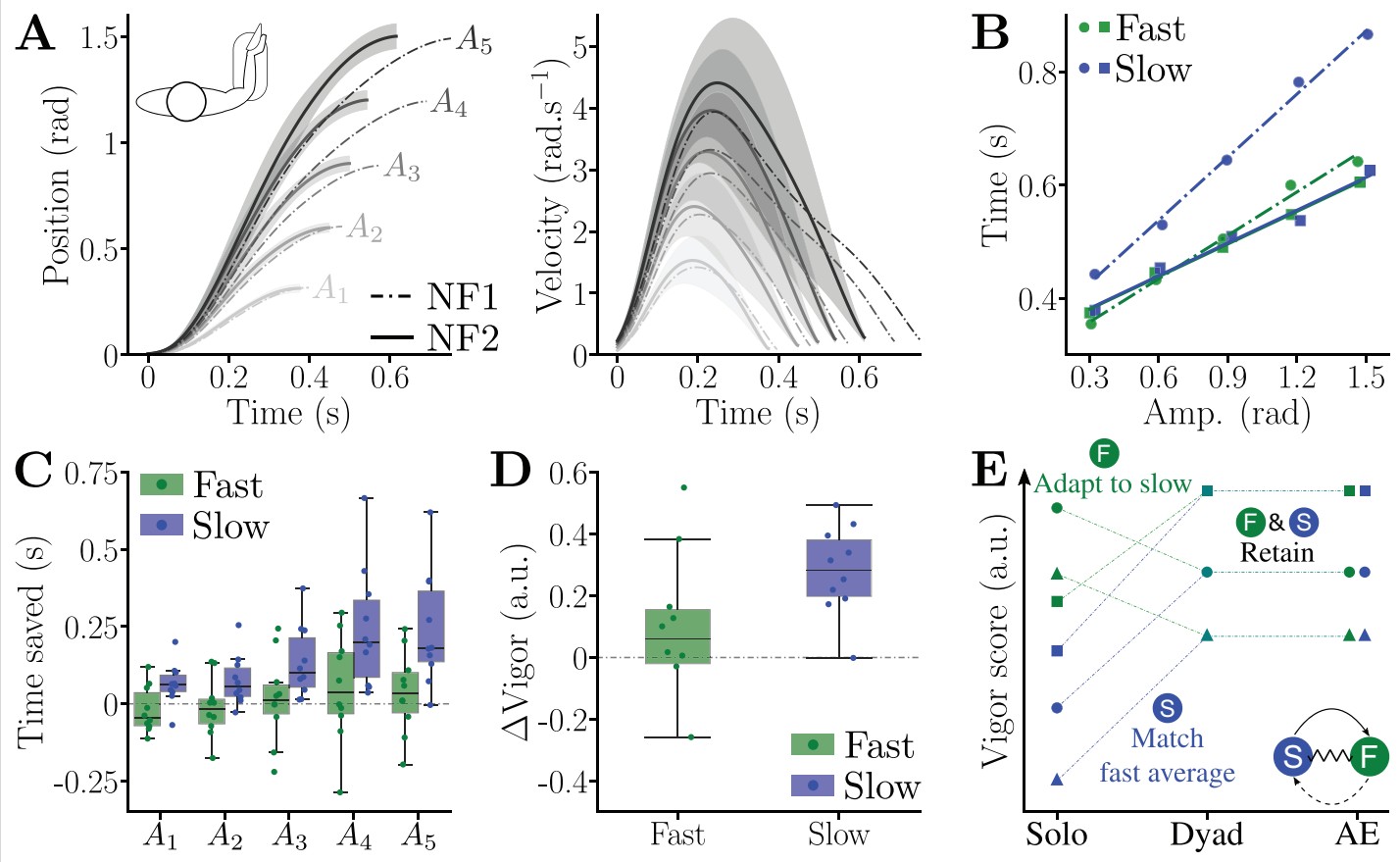

**Figure 5.** Kinematics and vigor before and after human–human interaction. Data before (NF1) and after (NF2) the dyadic session are represented using dashed-dotted and solid lines, respectively. (**A**) Population average of the position and velocity for each target. (**B**) Averaged amplitude–duration relationships for the fast and slow partners in NF1 and NF2. (**C**) Time saved between NF1 and NF2 for the fast and slow partners and for each target. (**D**) Difference in vigor scores between NF1 and NF2, for the fast and slow partners. (**E**) Illustration of the fast and slow partners' adaptations. When connected, the average vigor of the slow partners increases to match the one of the fast partners, while the fast partners' group changes its internal ranking in vigor to match the ranking of the slow partners. These adaptations are retained for both groups in after-effects (AE).

The online version of this article includes the following figure supplement(s) for figure 5:

**Figure supplement 1.** Effect of the connection on vigor and the affine amplitude–duration relationship.

effect of changing the cost of time of the fast partner, i.e. their vigor, on predicted movement duration (*Figure 4—figure supplement 3*). This analysis shows that modifying the fast partner's vigor only marginally impacts the optimal predicted time, with changes of 10 and 35 ms for a 30% increase in their cost of time for the $A_1$ and $A_5$ targets, respectively. These minor impacts are consistent with the fast partner's vigor not being a predictor of dyadic vigor. The *interactive adaptation* is therefore the only hypothesis that captures all the qualitative features of the experimental data, and the associated model provides accurate movement duration predictions, thereby complying with all the requested conditions for computational modeling summarized above. Furthermore, as the impact of the fast partner's vigor on the predicted time is small, the predicted motion plan can be considered as representing the dyad's emerging common plan, rather than a plan that would result from a fast leader that would just account for the slow partner's behavior.

## After effects of dyadic interaction

Did dyadic practice influence individual movement timing? To address this question, we analyzed how the movements of slow and faster partners changed with dyadic practice. *Figure 5A* shows that averaged trajectories across the population were notably different after (NF2) relative to before (NF1) practice. In particular, the average movement durations decreased, while the velocity profiles remained similar to those with both KL and KH, without terminal submovement. Furthermore, the

average amplitude–duration relationship in NF2 changed for the slow partners but not for the fast (*Figure 5B*).

A one-way ANOVA confirmed behavioral differences between groups and conditions (SS > 0.3, $p < 10^{-3}$, $\eta^2 > 0.08$ in both cases). In the first null-field block (NF1), the slow partners moved significantly slower than the fast partners ($p < 10^{-3}$, Cohen's $D = 0.72$) but not in the post-connection null-field block (NF2) ($p > 0.72$). Furthermore, the group of slow partners changed their movement durations between NF1 and NF2 ($p < 10^{-3}$, Cohen's $D = 0.8$) but not the group of fast partners ($p = 0.41$). Movement amplitude had a main effect on time saving ($W = 0.32$, $p < 10^{-3}$), which was confirmed across the population for targets $\{A_4, A_5\}$ ($p < 0.02$, Cohen's $D > 0.87$ in all cases, *Figure 5C*). The vigor analysis (*Figure 5D*, *Figure 5—figure supplement 1A*) confirmed this adaptation: while the slow partners' vigor increased between NF1 and NF2 ($p = 0.01$, Cohen's $D = 2.73$), the fast partners' vigor did not change ($p = 0.43$).

Interestingly, despite the change of motor plan of the slow partners, the NF1 and NF2 vigor scores over the whole population remained correlated ($p = 3.2 \cdot 10^{-3}$, $r = 0.62$). Furthermore, the vigors of the group of slow partners in NF2 were correlated to their behavior in NF1 ($p = 0.004$, $r = 0.87$). Surprisingly, the vigors of the group of fast partners in NF2 were correlated with the vigor of their slower partner in NF1 ($p = 0.034$, $r = 0.73$), but not with their own vigors in NF1 ($p = 0.14$, $r = 0.5$). This highlights a structural change in the repartition of vigor scores in the fast partners, reflecting their adaptation to the slow partners in each dyad.

Therefore, dyadic practice led to an adaptation of the average vigor in the slow partners, but their relative repartition was not modified. That is, fast participants within the group of slow partners remained fast after dyadic practice, and slow participants remained slow. In contrast, the average vigor scores of the fast partners remained unchanged in NF2 relative to NF1. However, there was a structural change in their repartition according to vigor, which matched the repartition of their slow partner after dyadic practice (*Figure 5E*). Importantly, at the dyad level, the adaptation is sometimes larger for the slow participant and sometimes larger for the fast participant. For instance, the dyad composed of subjects S5 and S6 shows a large decrease of vigor for the fast (S6) and an increase of a lower magnitude for the slow (S5) participant after connected practice (see *Figure 5—figure supplement 1* and *Appendix 2—table 1* for details).

## Discussion

The movements of individuals are characterized by their vigor, revealing how prone individuals are to perform actions fast or slowly in various conditions. Vigor is thought to be a fundamental idiosyncratic trait of volitional movements that relates to energy, time/reward, and performance/accuracy. How do individuals combine their vigor to coordinate movements when physically collaborating on joint actions? We addressed this question by investigating how partners connected by a virtual elastic band performed goal-directed hand movements without time constraints. Unlike previous human interaction studies, where timing was externally prescribed, either by a tracking task (*Groten et al., 2009*; *Melendez-Calderon et al., 2015*; *Ganesh et al., 2014*), explicit instructions (*Takagi et al., 2016*), or the introduction of an explicit reward when connected (*Reed and Peshkin, 2008*), our study allowed for natural and unbiased timing emergence.

We first observed that connected partners used a minimal interaction force to achieve synchronization, rapidly adapting within the initial trials and exhibiting negligible time differences in movement execution. This indicates that partners did not move independently, as proposed by *Takagi et al., 2016*, but instead rapidly adopted a common motor plan, likely facilitated by humans routinely interacting in daily tasks and the studies' task simplicity. Importantly, these findings show that sensory exchanges occur between partners even during simple point-to-point movements, not only in complex interaction scenarios as previously reported (*Ganesh et al., 2014*; *Sawers and Ting, 2014*; *Sawers et al., 2017*; *Sylos-Labini et al., 2018*). Second, in conditions without viscous load, dyadic movements were strikingly faster and smoother than the movements of either of their individuals, without compromising accuracy. With a load, the group of slower partners moved as fast as the faster partners, while the group of faster partners essentially maintained their pace. These findings confirm and extend the benefits of dyadic control, in terms of movement time and trajectory tracking accuracy previously observed in collaborative actions (*Reed and Peshkin, 2008*; *Ganesh et al., 2014*; *Takagi et al., 2017*).

Furthermore, dyads exhibited consistently slower or faster movement duration relative to the average over dyads, which was shown for different levels of effort and connection stiffness. As dyads' partners coordinated their actions with these systematic features, this allowed us to define *dyadic vigor* similarly to individual vigor (*Berret et al., 2018*; *Labaune et al., 2020*). Interestingly, we found no significant differences in reaction times between the groups of fast and slow partners, in contrast to previous works reporting a correlation between reaction time and vigor (*Reppert et al., 2018*). This discrepancy may stem from several factors. First, our task was not specifically designed to finely assess reaction times, e.g. using eye gaze motion initiation instead of hand motion initiation. Second, the correlation between these two parameters may be weak, as suggested by non-significant trends in previous works (*Choi et al., 2014*; *Verdel et al., 2023b*). Interestingly, we observed that dyadic practice led to significantly shorter reaction times, which may be due to increased attention (*Eason et al., 1969*), possibly related to task uncertainty, when one's motor behavior is directly impacted by a partner. This finding may have important implications for the design of robotic partners physically interacting with humans, where being synchronized with the user is critical.

We found a striking result regarding this dyadic vigor, which we expected to primarily correlate with the fast partners' vigor in a leader–follower scheme (*Losey et al., 2018*). While the connected participants had a movement duration close to that of the group of fast partners, dyadic vigor was instead predicted by the slower partner's vigor. This unexpected finding was consistently observed across varying stiffness and resistance conditions, where the slower partners' vigor was a significant predictor of dyadic vigor. This reorganization of the fast partners' vigor scores highlights the slow partners' pivotal role in determining the dyad's movement pace. While the slower partners move faster connected than alone, the faster partners adapt their effort accordingly to match the slow partners' influence. Note that the magnitude of the adaptation can be large for both of them, sometimes larger for the faster.

This finding challenges the leader–follower framework often used to describe human–human and human–robot interactions, where the leader would typically be the individual exerting greater motor force or being ahead in the movement (*Reed and Peshkin, 2008*; *Groten et al., 2009*; *Melendez-Calderon et al., 2015*). Our results show that this interpretation is misleading. It also challenges the weighted averaging hypothesis suggesting that the dyad vigor should be systematically in-between the slow and fast partners' initial vigor, with both contributing to its setting. Rather, our results demonstrate that a dyad motor plan emerges from a co-adaptation where the slower partner's vigor is the final determinant. Instead of a fixed leader–follower relationship or weighted adaptation, human–human interactions appear to depend on the context (e.g. the available information to each partner, or the type of task) rather than on predefined roles or a simple mix of the initial individual strategies (*Börner et al., 2023*). This should be considered when designing human–robot interaction controllers (*Li et al., 2022*), contrary to the numerous control methods relying on a leader–follower strategy to provide assistance (*Losey et al., 2018*).

Following connected training, strong after effects were observed in both fast and slow partners. For slow partners, a large increase in the individual vigor was observed. Interestingly, the vigor scores of the fast partners were reorganized depending on their partner, while the order of the slow partners' scores was unchanged by dyadic practice, similarly to dyadic vigor during the connected session. This highlights how both partners adapted. The slow partners increased vigor, converging to the average vigor of the fast partners, while the fast partners reorganized their vigor score corresponding to the repartition of the slow partners. These effects persisted across 100 simple reaching movements, whereas after-effects of force fields or visuomotor rotations are typically re-optimized within a few trials (*Herzfeld et al., 2014*; *Verdel et al., 2023a*), and effects of previous attempts at modifying individual vigor tended to vanish through unconstrained practice (*Mazzoni et al., 2012*). This persistence indicates a potential change in the time perception for both groups after practicing with a partner.

These findings suggest joint training as a potential therapeutic approach to restore declining movement vigor, such as in individuals with Parkinson's disease (*Mazzoni et al., 2007*; *Panigrahi et al., 2015*). In this context, future works should examine whether the vigor changes observed after dyadic practice are retained over longer periods or with intensive movement sessions, or whether they end up vanishing as was observed in studies where participants were instructed to move faster using visual cues (*Mazzoni et al., 2012*). If the vigor adaptations observed during and after physical human–human interactions such as in the present paper were shown to persist over long periods, this

would challenge the view that vigor is a strictly stable, idiosyncratic trait. Instead, it would suggest that the exploration of other motor solutions and rewards can modify the optimal tradeoff selected by the CNS. Such fundamental changes in vigor could potentially be assessed by analyzing the evolution of dopamine levels in the basal ganglia (*Dudman and Gerfen, 2015*; *Dudman and Krakauer, 2016*) (e.g. using positron emission tomography; *Elsinga et al., 2006*), as dopamine plays a major role in vigor regulation (*Kawagoe et al., 1998*; *Salamone et al., 2009*; *Nicola, 2010*; *Tachibana and Hikosaka, 2012*; *Rueda-Orozco and Robbe, 2015*; *Jurado-Parras et al., 2020*).

To interpret the mechanism behind the coordinated movement timing and trajectory in dyads, we developed a computational model based on stochastic nonlinear optimal control with free final time, thereby combining several computational approaches (*Berret and Baud-Bovy, 2022*; *Verdel et al., 2023b*; *Berret et al., 2024*), in which dyads plan movements based on the slower partner's average vigor under uncertainty, as was observed during connected movements. This model successfully predicts the key qualitative features of our data, in contrast to hypotheses that assume (1) independent motor plans for slow and fast partners (*co-activity*), (2) a vigor distribution decided by a leader and accepted by a follower (*leader–follower*), or (3) a negotiation based on a weighting of their initial strategies in the task (*weighted adaptation*). These results further show the role of the slow partner and the critical impact of uncertainty with respect to the partner's plan in the emergence of dyadic vigor. These findings argue in favor of a dynamic adaptation of an individual's motor strategy caused by the presence of the other partner (and its inherent induced uncertainty).

Focusing on dyadic movement timing, the proposed model necessarily involved several simplifying assumptions. First, because current control theory does not, to our knowledge, yet allow the simulation of multiple physically interacting agents with nonlinear and uncertain dynamics under free-final-time conditions, dyadic interaction was approximated using a single-agent formulation. Second, we modeled the *average* dyad rather than generating dyad-specific predictions, as individual model fitting was unlikely to yield additional insights into the underlying coordination principles. We further restricted the model to the open-loop component of motor control, consistent with the stable, reproducible behavior observed after an initial increase in interaction force. As we predicted, averaged movement durations across the experimental population for each condition, the influence of feedback control (*Todorov and Jordan, 2002*) was minimal. Extending the model with feedback control would require exploring the interplay between feedback control and (re)planning mechanisms that are beyond the scope of the present study, which primarily aims to investigate the role of individual vigor in shaping human–human interactions.

More fundamentally, vigor is thought to emerge from the interplay between effort, accuracy, and time or reward. Inter-individual differences in vigor may therefore reflect differences in sensitivity to any of these factors (*Shadmehr and Ahmed, 2020*). However, disentangling these sensitivities based solely on behavioral data is difficult. Here, we estimated the cost of time for the average participant using a previously established deterministic effort model with fixed final time (*Berret and Jean, 2016*). Because the model does not explicitly account for accuracy (e.g. through variance minimization), its generalizability to tasks with different accuracy constraints is limited.

Our model achieved minimal movement duration prediction error across all 30 tested conditions, covering five amplitudes and six load conditions. Furthermore, we conservatively assumed that participants used a single cost function across all conditions, even though humans can adjust strategies based on task demands (*Poirier et al., 2024*). A sensitivity analysis confirmed that increasing the slow partner's average movement duration leads to an increase in the optimal dyadic movement duration, while changes in the fast partner's cost of time had marginal influence, consistent with the observed correlation between slower partners' vigor and dyadic vigor.

These results indicate that the faster individual in a dyad relies on an internal estimate of the partner's movements distribution, providing a plausible mechanism for the faster dyadic movements in the absence of viscous load, the preserved or decreased average movement duration of the fast partners in the presence of viscous load, and the dependence of dyadic vigor on the slower partner. This model offers a principled basis for developing robotic systems that interact with humans similar to a human partner, with applications in collaborative robots for manufacturing (*Liu et al., 2024*), rehabilitation robotics (*Mehrholz et al., 2020*), exoskeletons (*Dalla Gasperina et al., 2021*) and supernumerary robotic limbs (*Eden et al., 2022*).

# Methods

## Participants and experimental setup

The experimental protocol was approved by the Imperial College's Science, Engineering and Technology Research Ethics Committee (SETREC number 7112072). $N = 20$ right-handed young adults (4 females) with no known motor or cognitive impairment participated in the experiment in 10 dyads of two partners. Their biographical data were: age $27.5 \pm 5.5$ years old, weight $73.7 \pm 14.5$ kg, height $175 \pm 8.5$ cm and right hand length $19.1 \pm 1.2$ cm (from the wrist flexion/extension rotation axis to the farthest fingertip). Each participant was informed about the experiment and signed an informed consent form before participating in it.

Each of the dyad's partners interacted with an HRX-1 robotic interface (HumanRobotiX, London, UK) using their right wrists, while seated in front of their respective monitor (**Figure 1B**). Participants had to control a 0.6-cm diameter cursor toward a $2.7 \times 2.7$ cm$^2$ target via wrist flexion/extension. These dimensions ensured consistent movement amplitudes across participants with minimal accuracy demands. The two HRX-1 were controlled and synchronized at $1$ kHz using a TI PiccoloF28069M LaunchPad. Wrist flexion/extension angles were recorded at $300$ Hz with an optical encoder (MILE 512-6400, 6400 counts per turn). Angles (18°, 36°, 54°, 72°, 90°) were linearly mapped to targets $(A_1, A_2, A_3, A_4, A_5)$ on the monitor, with 9.3 cm between the centers of consecutive targets (**Figure 1B**).

## Experimental protocols

Participants completed two experimental sessions (**Figure 1C**):

- In a first *solo session*, the two participants performed goal-directed wrist flexion/extension movements independently. These movements were used to (1) verify that our population sample followed a vigor law, (2) estimate their individual vigor and cost of time for computational modeling, and (3) establish a baseline to analyze after-effects of the human–human interaction.
- In the subsequent *dyadic session*, they were coupled via a virtual elastic band implemented through the robotic interfaces. The data enabled us to investigate movement coordination between the partners and its dependence on individual vigor.

The participants performed back and forth reaching at preferred speed, starting with a wrist extension movement from $A_0$ to one of the targets $\{A_1, A_2, A_3, A_4, A_5\}$, followed by a flexion movement, back to $A_0$. After each movement, the timing of the next movement start was drawn from a uniform distribution to prevent anticipation and rhythmic behavior. After holding at $A_0$ for $2 \pm 0.15$ s, $A_0$ was switched off and a target was switched on for $2 \pm 0.15$ s, then $A_0$ was switched on again. This cycle was then repeated.

### Solo session

Participants completed 3 blocks of 100 trials (10 extension and 10 flexion movements per amplitude: $\{18°, 36°, 54°, 72°, 90°\}$) in a fixed pseudo-random sequence. A 2-min break was provided between blocks to prevent fatigue. The first block was performed in 'transparent' mode (robot motor off). The next two blocks introduced resistive viscous loads ($\nu \in \{0.075, 0.15\}$ Nm s/rad in random order) to examine vigor robustness under varying effort costs. A final washout block was introduced to avoid carrying after-effects of the two visco-resistive blocks into the connected session. This block was shorter than the other blocks, consisting of 40 reaching movements with 4 flexions and 4 extensions per amplitude. The same amplitudes and targets were used as for the other blocks, but their order was not randomized, i.e. the blocks consisted of four series of movements toward $A_1$, then $A_2$, …, then $A_5$.

### Dyadic session

Dyads of participants completed 6 blocks of 100 trials while coupled via a virtual visco-elastic band. The first two blocks used an elastic band of stiffness $\kappa \in \{0.5, 1.6\}$ Nm/rad in randomized order. The next four blocks tested all $\kappa$ and $\nu$ combinations in random order to explore time–effort tradeoffs. A seventh block without coupling (i.e. $\kappa = \nu \triangleq 0$) was used to assess carryover effects of the interaction with the partner on individual vigor. Participants were informed that the robot would provide assistance but were not told that they were interacting with another human. When debriefed at the end of

the experiment, only one of the 20 participants reported having realized that they were connected to a partner. Most participants believed that they were interacting with a version of themselves or with a robot exhibiting stochastic behavior.

## Data processing

### Wrist kinematics

The wrist velocity and acceleration were computed by numerical differentiation, after low pass filtering of position data using a forward–backward fifth order Butterworth filter with 5 Hz cutoff frequency. Wrist flexion and extension movements were first grossly segmented based on the timings of the targets, then a threshold of 5% of peak velocity was used to compute the start and end of each movement. Leftward and rightward movements were pooled together as their vigor was similar, consistently with expectations (*Labaune et al., 2020*; *Verdel et al., 2023b*).

The amplitude–duration relationship of each participant was obtained by (1) averaging movement durations and amplitudes per target and (2) performing an affine least-square regression (using the *LinearRegression* function of *scikit-learn.linear_model*). The amplitude–duration relationship of a group of participants was obtained by (1) averaging across the group the average movements durations and amplitudes of each participant, for each target separately, then (2) performing an affine regression.

All the movement durations were obtained from affine fittings of the amplitude–duration relationships. The vigor scores in the VL and VH conditions were computed using the average movement duration of the population for each target $\bar{T}(A_k)$ from NF1, which allowed us to illustrate the general decrease in vigor with a resistive torque. The same approach was adopted for the dyadic session, using the population-averaged movement duration in KL and KH to compute the vigor scores in {KLVL, KLVH} and {KHVL, KHVH}, respectively. As a consequence, the vigor scores were equally distributed around 1 only in the NF1, KL, and KH conditions. To analyze the variance, the vigor scores were computed based on the average movement duration of the population in each condition.

The reaction time of participants was computed as the time difference between the apparition of a target and the first instant where the wrist angular velocity exceeded $0.01\,\text{rad/s}$.

### Interaction forces

The reported average values were obtained by (1) computing the absolute value of the interaction efforts as a function of time for each whole movement, and (2) computing the average value of this absolute profile.

## Statistical analysis

The statistical analyses were performed using custom Python 3.8 scripts, the *statsmodel* package (*Seabold and Perktold, 2010*), and the *Pingouin* package (*Vallat, 2018*). For all metrics, the results were first averaged in each participant, condition, and movement amplitude before performing statistical analyses. Normality (*Shapiro–Wilk* test; *Shapiro and Wilk, 1965*) and sphericity (*Mauchly's* test; *Mauchly, 1940*) of the distribution of the residuals were not always verified. Therefore, for convenience, all within-subjects and within-dyads main effects of the condition and amplitude were assessed using Friedman tests, while unpaired effects between groups were assessed using Kruskal–Wallis tests, with a significance level set at 5%.

In case of a significant main effect, post hoc pairwise comparisons were performed using Wilcoxon tests within subjects and Mann–Whitney tests between the fast and slow partners. The significance level of post hoc comparisons was set at 5%, with a Bonferroni–Holm correction for multiple comparisons. All the significant comparisons are reported with the Cohen's $D$ value to illustrate the effect size, where $D < 0.4$ would be considered as small and reported as a weak result.

The proportion of variance due to inter-individual variability was quantified using a variance decomposition of the recorded movement durations data. Sum-of-square components were used to compute the percentage of variance explained by inter-individual variability as follows:

$$\text{Var}_{\text{inter}} = 100\frac{\text{SS}_{\text{inter}}}{\text{SS}_{\text{inter}} + \text{SS}_{\text{intra}}}. \tag{4}$$

The prediction of the dyads' vigor in the connected conditions was performed using an LMM attempting to correlate the vigor of the dyad to the vigor of its fastest and slowest participants as

$$v^d \sim \gamma^f v^f + \gamma^s v^s, \tag{5}$$

where $v^d$ is the vigor of dyads in one of the connected conditions $\{KL, KLVL, KLVH, KH, KHVL, KHVH\}$, $v^f$ is the vigor of the dyad's faster partner in the NF1 condition, and $v^s$ of the slower partner. Throughout the paper, a Bonferroni–Holm correction was applied for multiple regressions.

## Computational modeling and simulations

### Solo session

Data collected in solo setting were used to model the dyad behavior and predict the dyadic behavior from individual movement characteristics. The solo session included three levels of viscous resistance $\nu \in \{0, 0.075, 0.15\}$ (referred to as NF1, VL, and VH, respectively), resulting in the task dynamics

$$I\ddot{q} = \tau - D\dot{q} - \nu\dot{q}, \tag{6}$$

where $\ddot{q}$ [rad/s$^2$] is the wrist angular acceleration, $\dot{q}$ [rad/s] the angular velocity, $\tau$ [Nm] the human wrist torque, $I = 9.2 \cdot 10^{-3}$ Nm s$^2$ the population average hand + robot inertia, and $D = 0.03$ Nm s is the wrist damping selected in the range of values identified during null-field movements in *Gielen and Houk, 1984*; *Milner and Cloutier, 1998*.

Following the inverse optimal control procedure of *Berret and Jean, 2016*; *Verdel et al., 2023b*, we first estimated the cost of time of the average participant based on the population average amplitude–duration relationship (see *Figure 2B*) in NF1 without viscous load. The estimated amplitude–duration relationship for the population was

$$\overline{T}(A_k) = 0.314 A_k + 0.298, \quad A_k \in \{18°, 36°, 54°, 72°, 90°\}. \tag{7}$$

Moving the hand from an initial angle $q_0$ to an end angle $q_e$ was formulated as an optimal control problem, using the minimum torque change framework (*Uno et al., 1989*). Utilizing the state $\mathbf{x}_k = (q_k, \dot{q}_k, \tau_k)$ and the motor command $u_k = \dot{\tau}_k$ (as in *Verdel et al., 2023b*), the cost of effort to minimize for amplitude $k$ was

$$E_k(u_k) = \int_0^{\overline{T}(A_k)} [\mathbf{x}_e - \mathbf{x}_k(t)]' Q_k [\mathbf{x}_e - \mathbf{x}_k(t)] + \beta u_k^2 \, dt, \tag{8}$$

where $\mathbf{x}_e = [q_e, 0]'$ is the final state, $\overline{T}(A_k)$ is the final time of the average participant for amplitude $k$ computed from *Equation 7*. $Q_k = \mathrm{diag}(15.5/A_k, 0, 0)$ and $\beta = 0.95$, determined by trial and error, allow adjusting the compromise between minimizing the torque change and the distance to the target. As detailed in *Berret and Jean, 2016*, the human cost of time $G(t)$ can be identified by fitting and integrating a sigmoid curve on the Hamiltonians of the fixed final time problems from *Equation 8*, yielding

$$G(t) = p_1 \left( 1 - \left[ 1 + (t/p_3)^{p_2} \right]^{-p_4} \right), \tag{9}$$

where $\mathbf{p} = (p_1, p_2, p_3, p_4) = (1.5713, 6.9723, 0.4497, 1)$ was identified for the average participant in our sample. Using the same effort formulation as in *Equation 8*, it is then possible to predict the movement duration and cost of effort for $\nu \in \{0.075, 0.15\}$ Nm s/rad by minimizing

$$V_k(u_k, T_k) = G(T_k) + \int_0^{T_k} [\mathbf{x}_e - \mathbf{x}_k(t)]' Q_k [\mathbf{x}_e - \mathbf{x}_k(t)] + \beta u_k^2 \, dt, \tag{10}$$

in free final time, i.e. when the movement duration is optimized simultaneously to the effort cost. Here, $V_k$ is the cost to minimize for amplitude $A_k$ and $T_k \in [0.01, 10]$ s the optimal final time for amplitude $k$. Note that minimizing $V_k$ with $\nu \triangleq 0$ leads to $T_k = \overline{T}(A_k)$, and to the same predicted motor behaviors as the fixed final time optimal control problems minimizing *Equation 8*.

The simulations of the solo session were performed using the Matlab (R2021a, MathWorks) version of GPOPS-II (*Benson et al., 2006*; *Garg et al., 2010*; *Rao et al., 2010*), which is based on an orthogonal collocation method relying on SNOPT to solve the nonlinear programming problem (*Gill et al., 2005*).

## Dyadic session

The task of the dyadic session included two levels of connection stiffness $\kappa \in \{0.5, 1.6\}$ Nm/rad, and three levels of viscous resistance as for the solo session, yielding the coupled dynamics of the fast and slow partners

$$\begin{cases} I\ddot{q}^f = \tau^f - D\dot{q}^f - \kappa(q^f - q^s) - \nu\dot{q}^f \\ I\ddot{q}^s = \tau^s - D\dot{q}^s - \kappa(q^s - q^f) - \nu\dot{q}^s \end{cases}, \tag{11}$$

where the superscripts $f$ and $s$ indicate the fast and slow participants, respectively, $\tau^f$ and $\tau^s$ are their wrist torques, $I$ is the total inertia of the wrist + exoskeleton, $D$ the total damping as in the solo session, and $\ddot{q}$, $\dot{q}$, and $q$ are the wrist angular acceleration, speed, and position, respectively.

To simulate movements according to the *co-activity* hypothesis, we split the participants into the groups of the faster and slower partners in each dyad according to their vigor in NF1. Furthermore, to simulate the *interactive adaptation* hypothesis, assuming a coordination of the partners' motor plans, we had to identify the cost of time of the fast and slow partners, which was done based on their respective averaged amplitude–duration relationships

$$\begin{cases} \overline{T}^f(A_k) = 0.235A_k + 0.285 \\ \overline{T}^s(A_k) = 0.35A_k + 0.323 \end{cases} \tag{12}$$

Importantly, to simulate *interactive adaptation* we identified the cost of time of the fast ($\mathbf{p} = (3.9615, 5.8958, 0.4459, 1)$) and the slow partners ($\mathbf{p} = (1.0184 \cdot 10^7, 11.2958, 1.9980, 1.4447 \cdot 10^{-5})$) separately, which allowed us to simulate the effects of the average partners' individual vigor on the dyad's motor plan (see *Equation 9* for the definition). Then, the simulated hypotheses induced different state variables, conditions, and numerical methods to predict movements under the coupled dynamics as described below.

### Co-activity

Co-activity assumes that the participants do not coordinate their movements, with each partner trying to perform their predefined desired trajectory (as suggested in *Takagi et al., 2016*). For the sake of simplicity, we set these predefined plans using simple minimum jerk plans with durations identified in the lone session for the fast and slow participant separately. Optimal control commands $u_k^f(t)$ and $u_k^s(t)$ to track the obtained desired trajectories for the fast and slow partners for target $k$, respectively, are computed using a linear quadratic regulator. We use the state $\mathbf{x} = (q_k^f, \dot{q}_k^f, q_k^s, \dot{q}_k^s)$ and the cost matrices $\mathbf{Q} = \text{diag}(100, 0.1, 100, 0.1)$ and $\mathbf{R} = \mathbb{1}_2$. The fast partners' trajectory is extended with null values to match the duration of the slow partners' plan. The resulting coupled movements are Euler integrated using these control inputs in *Equation 11*.

### Interactive adaptation

Here, we assume that dyads' partners collaboratively optimize their movements based on the solo behavior of the slow partner, who 'drives' the co-adaptation. Our model describes the dyad's motion plan using a single cost function. While one would intuitively seek multiple agents modeling, to our knowledge, there is no tractable solution to simulate multiple agents under uncertainty, in free final time, and not based on usual quadratic cost functions. Both partners contribute to selecting the final movement control by optimizing the common motor plan based on a prediction of the slow partner's motion, which is uncertain for the fast partner. For the sake of simplicity, we represent the slow partner's trajectory as a distribution of minimal jerk trajectories (*Hogan, 1984*)

$$q_k^s(\xi_k) = q_0 + A_k \zeta^3(10 - 15\zeta + 6\zeta^2), \quad \zeta \equiv \frac{t}{\xi_k}, \tag{13}$$

where the movement duration $T_k^s = \xi_k$ is a random variable with mean $\mu_k$ and standard deviation $\sigma_k$. Note that $q_0$ and the $\{A_k\}$ are considered common and known to both partners, and no assumption is made on $\xi_k$'s distribution beyond its deviation. The task dynamics inferred by the fast partner are

$$I\ddot{q}_k^f = \tau_k^f - (D + \nu)\dot{q}_k^f - \tau_k^i, \quad \tau_k^i = \kappa[q_k^f - q_k^s(\xi_k)], \tag{14}$$

where $\tau_k^i$ is the interaction torque for target $k$. Since $\xi_k$ is a random variable, the uncertain interaction torque change can be approximated based on a linearization of the velocity of the slow partner around $\mu_k$:

$$\dot{q}_k^s(\xi_k) = \dot{q}_k^s(\mu_k) + \left.\frac{\partial \dot{q}_k^s}{\partial \xi_k}\right|_{\mu_k}(\xi_k - \mu_k), \tag{15}$$

yielding

$$\dot{\tau}_k^i(\xi_k) = \kappa\left[\dot{q}_k^f - \dot{q}_k^s(\mu_k)\right] - \kappa\left.\frac{\partial \dot{q}_k^s}{\partial \xi_k}\right|_{\mu_k}(\xi_k - \mu_k). \tag{16}$$

The expected cost function of the dyad can then be expressed as a tradeoff between the fast partner's cost function when moving alone (including their cost of time) and the uncertain interaction torque change with respect to the behavior of the slow participant, reflecting a strategy of stabilization of the interaction efforts. We consider that participants would minimize a cost related to the expected interaction torque and its uncertainty, while neglecting the cross-covariance terms, expressed as in the main text:

$$C_k(u_k, T_k) = \mathbb{E}\left[\underbrace{V_k^f(u_k, T_k)}_{\substack{\text{Alone cost}\\\text{fast}}} + Q_\tau \int_0^{T_k}\left(\underbrace{\kappa^2\left[\dot{q}_k^f - \dot{q}_k^s(\mu_k)\right]^2}_{\substack{\text{Expected interaction}\\\text{torque variations}}} + \underbrace{\kappa^2\left[\left.\frac{\partial \dot{q}_k^s}{\partial \xi_k}\right|_{\mu_k}(\xi_k - \mu_k)\right]^2}_{\text{Uncertainty term}}\right)dt\right]. \tag{17}$$

This leads to the expected cost

$$C_k(u_k, T_k) = \overline{V}_k^f(u_k, T_k) + Q_\tau \kappa^2 \int_0^{T_k}\left(\left[\bar{\dot{q}}_k^f - \dot{q}_k^s(\mu_k)\right]^2 + P_{\dot{q}}^f + \left[\left.\frac{\partial \dot{q}_k^s}{\partial \xi_k}\right|_{\mu_k}\right]^2 \sigma_k^2\right)dt, \tag{18}$$

where $\overline{V}_k^f = \mathbb{E}[V_k^f]$, $u_k$ is the dyadic motor command, $\sigma_k^2$ is the variance of $\xi_k$ and $P_{\dot{q}}^f$ the velocity component of the state covariance matrix. This stochastic optimization problem can be approximated under a deterministic form and solved within the stochastic optimal open-loop control framework of *Berret et al., 2024* (see in particular Equation 3), which provided codes on which we based our simulations, using Julia (*Bezanson et al., 2017*), JuMP (*Lubin et al., 2023*), and Ipopt (*Wächter and Biegler, 2006*), where the problem was transcribed into an NLP problem using a trapezoidal scheme (*Betts, 2010*).

In the simulation results of *Figure 4B* corresponding to the *interactive adaptation* hypothesis, the dyads were considered to use the same cost function for all amplitudes and conditions with $Q_\tau = 3$. The uncertainty with respect to the slow partner's behavior was assumed to increase with movement amplitude and resistive viscosity. Specifically, we used values uniformly spread within $\sigma_k \in [0.15, 0.25]\,\text{s}$ for KL and KH, within $\sigma_k \in [0.2, 0.3]\,\text{s}$ for KLVL and KHVL, and within $\sigma_k \in [0.32, 0.4]\,\text{s}$ for KLVH and KHVH.

## Acknowledgements

We thank Jingwen Zhao for her help during the development of the experimental setup, Claudia Clopath for her com ments on a prior version of the manuscript and Dario Farina for his help in funding administration. This study was funded in part by the European Commission (FETOPEN H2020 899626 NIMA, ICT 871767 ReHyb and ERC NaturalBionics).

# Additional information

## Funding

| Funder | Grant reference number | Author |
|---|---|---|
| European Research Council | 810346 | Etienne Burdet |
| H2020 Future and Emerging Technologies | 899626 | Etienne Burdet |
| H2020 LEIT Information and Communication Technologies | 871767 | Etienne Burdet |

The funders had no role in study design, data collection, and interpretation, or the decision to submit the work for publication.

## Author contributions

Dorian Verdel, Conceptualization, Data curation, Software, Formal analysis, Validation, Investigation, Visualization, Methodology, Writing – original draft, Writing – review and editing; Bastien Berret, Conceptualization, Software, Formal analysis, Methodology, Writing – review and editing; Etienne Burdet, Conceptualization, Resources, Supervision, Funding acquisition, Visualization, Methodology, Project administration, Writing – review and editing

## Author ORCIDs

Dorian Verdel  https://orcid.org/0000-0001-7307-1086
Bastien Berret  https://orcid.org/0000-0002-0779-7724
Etienne Burdet  https://orcid.org/0000-0002-2123-0185

## Ethics

The experimental protocol was approved by the Imperial College's Science, Engineering and Technology Research Ethics Committee (SETREC number 7112072). N = 20 right-handed young adults (4 females) with no known motor or cognitive impairment participated in the experiment in 10 dyads of two partners. Each participant was informed about the experiment and signed an informed consent form before participating in it.

Reviewer #1 (Public review): https://doi.org/10.7554/eLife.109781.3.sa1
Reviewer #2 (Public review): https://doi.org/10.7554/eLife.109781.3.sa2
Reviewer #3 (Public review): https://doi.org/10.7554/eLife.109781.3.sa3
Author response https://doi.org/10.7554/eLife.109781.3.sa4

# Additional files

## Supplementary files

MDAR checklist

## Data availability

All data supporting the findings of this study are available within the paper, the supplementary material, and are publicly available on Zenodo at https://doi.org/10.5281/zenodo.17169711.

The following dataset was generated:

| Author(s) | Year | Dataset title | Dataset URL | Database and Identifier |
|---|---|---|---|---|
| Verdel D, Berret B, Burdet E | 2025 | A dataset of wrist reaching movements for the study of collaborative human motor control | https://doi.org/10.5281/zenodo.17169711 | Zenodo, 10.5281/zenodo.17169711 |

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

# Appendix 1

## Performance

As we have shown when analyzing the movement duration of dyads, there was an increase in time performance for KL and KH, which was not observed when an additional viscous effort was imposed by the exoskeletons. However, it is yet unclear whether this reduction of movement time comes at the cost of a reduced accuracy, as would predict Fitts' law (*Fitts, 1954*), and whether it would impact other objective metrics of human movement performance such as smoothness (*Balasubramanian et al., 2015*). Therefore, we conducted an analysis of movement smoothness and an analysis of accuracy. Accuracy was estimated by computing the standard deviation of the wrist joint position after the end of the main movement (see Methods for the segmentation of movements). The results of these analyses are summarized in *Figure 3—figure supplement 3* alongside the saved time when compared to NF1.

### Movement smoothness

As highlighted by *Figure 3—figure supplement 3*, there is a trend of improved movement smoothness with both used metrics and for all conditions. Interestingly, a main effect of the human–human connection on movement smoothness was observed for all conditions, whether with passive exoskeletons (for both metrics: $W > .52$, $p < 10^{-3}$), with VL (for both metrics: $W > 0.23$, $p < 0.01$), or with VH (for the dimensionless jerk: $W = 0.23$, $p = 0.01$).

For the passive exoskeleton conditions (NF1, KL, and KH), the connection induced significantly smoother movements in KL and KH than in NF1 according to both metrics (in all cases: $p < 2 \cdot 10^{-3}$, Cohen's $D > 0.83$), except SPARC for the comparison between NF1 and KL. Furthermore, movements were smoother in KH than in KL (for both metrics: $p < 2 \cdot 10^{-3}$, Cohen's $D > 0.58$).

For the low viscous resistance conditions (VL, KLVL, and KHVL), the connection induced significantly smoother movements in KLVL and KHVL than in VL according to both metrics (in all cases: $p < 6 \cdot 10^{-3}$, Cohen's $D > 0.61$). No difference was observed between KLVL and KHVL.

For the high viscous resistance conditions (VH, KLVH, and KHVH), the connection induced significantly smoother movements in KLVH and KHVH than in VH according to the dimensionless jerk (in all cases: $p < 0.025$, Cohen's $D > 0.49$). No difference was observed between KLVH and KHVH.

In sum, our analysis showed a clear improvement of movement smoothness for both KL and KH compared to when participants are moving alone. This result was shown to hold throughout five movement amplitudes and for three different task dynamics.

### Accuracy

As highlighted by *Figure 3—figure supplement 3*, the reported reduction of movement time and increased smoothness did not seem to come at the expense of accuracy. In fact, the connection between participants had no significant effect on movement accuracy.

In sum, overall movement quality – in terms of time, smoothness, and accuracy – was either improved or not impacted (for the accuracy mainly) by the connection between participants. Importantly, this result was shown to hold for two levels of stiffness, throughout five amplitudes, and throughout three task dynamics. It can also be noted that movement accuracy was improved by the introduction of a viscous resistance in the dynamics of the exoskeletons, although the results, out of our scope of analysis, are not detailed here.

# Appendix 2

**Appendix 2—table 1.** Summary of the vigor scores of participants through the experiment. The vigor scores in the VH and VL conditions (with and without connection) were computed using the average movement durations collected in the corresponding condition without viscous resistance to better show the effects of changes in the dynamics. Vigor scores in the NF2 condition were computed using average durations from the NF1 conditions to highlight changes after exposure to the human–human connection.

| Dyad Id. | Subject Id. | Solo | | | Dyads | | | | | | AE |
|---|---|---|---|---|---|---|---|---|---|---|---|
| | | | | | High stiffness | | | Low stiffness | | | |
| | | NF1 | VL | VH | KH | VL | VH | KL | VL | VH | NF2 |
| | S1 | 1.4 | 1.34 | 1.19 | | | | | | | 1.42 |
| D1 | S2 | 1.47 | 1.12 | 1.04 | 1.44 | 0.86 | 0.68 | 1.32 | 0.93 | 0.72 | 1.6 |
| | S3 | 1.37 | 1.15 | 1.17 | | | | | | | 1.11 |
| D2 | S4 | 1 | 0.78 | 0.64 | 1.21 | 0.82 | 0.64 | 1.17 | 0.79 | 0.64 | 1 |
| | S5 | 0.75 | 0.66 | 0.42 | | | | | | | 1.18 |
| D3 | S6 | 1.58 | 1.19 | 0.94 | 0.84 | 0.69 | 0.59 | 0.97 | 0.75 | 0.54 | 1.05 |
| | S7 | 0.8 | 0.67 | 0.61 | | | | | | | 0.99 |
| D4 | S8 | 0.91 | 0.84 | 0.8 | 0.94 | 0.64 | 0.56 | 0.87 | 0.64 | 0.57 | 1.29 |
| | S9 | 0.91 | 0.82 | 0.81 | | | | | | | 1.17 |
| D5 | S10 | 1.05 | 1.16 | 0.54 | 1.1 | 0.7 | 0.55 | 1.06 | 0.75 | 0.61 | 1.6 |
| | S11 | 1.74 | 1.28 | 1.02 | | | | | | | 1.87 |
| D6 | S12 | 1.41 | 1.11 | 0.92 | 1.32 | 0.85 | 0.72 | 1.38 | 0.92 | 0.76 | 1.75 |
| | S13 | 1.02 | 0.7 | 0.56 | | | | | | | 1.18 |
| D7 | S14 | 0.89 | 0.64 | 0.52 | 0.94 | 0.59 | 0.43 | 1.06 | 0.64 | 0.47 | 1.06 |
| | S15 | 1.11 | 1.01 | 0.82 | | | | | | | 1.21 |
| D8 | S16 | 0.95 | 0.98 | 0.75 | 1.03 | 0.61 | 0.57 | 1.1 | 0.7 | 0.57 | 1.35 |
| | S17 | 0.86 | 0.83 | 0.79 | | | | | | | 0.83 |
| D9 | S18 | 0.74 | 0.8 | 0.93 | 0.79 | 0.66 | 0.61 | 0.83 | 0.67 | 0.6 | 1.05 |
| | S19 | 1.01 | 0.73 | 0.65 | | | | | | | 1.01 |
| D10 | S20 | 0.61 | 0.71 | 0.56 | 0.8 | 0.56 | 0.49 | 0.68 | 0.53 | 0.59 | 1.1 |

