## [Editor Report · eLife Assessment]

This is an **important** study showing that movement vigor is not solely an individual property but emerges through interaction when two people are physically linked. The evidence is **convincing**, supported by a well-controlled experimental design and modeling that closely match the observed behavior. While the authors provided a helpful comparison of several candidate models of human-human interaction dynamics, the statistical power remains limited.

---

## [Referee Report · Reviewer #1 (Public review)]

Summary:

The authors present a novel investigation of movement vigor of individuals completing a synchronous extension-flexion task. Participants were placed into groups of two (so-called "dyads") and asked to complete shared movements (connected via a virtual loaded spring) to targets placed at varying amplitudes. The authors attempted to quantify what, if any, adjustments in movement vigor individual participants made during the dyadic movements, given the combined or co-dependent nature of the task. This is a novel, timely question of interest within the broader field of human sensorimotor control.

Participants from each dyad were labeled as "slow" (low vigor) or "fast" (high vigor), and their respective contributions to the combined movement metrics assessed. The authors presented four candidate models for dyad interactions: (a) independent motor plans (i.e., co-activity hypothesis), (b) individual-led motor plans (i.e., leader-follower hypothesis), (c) generalization to a weighted average motor plan (i.e., weighted adaptation hypothesis), and (d) an uncertainty-based model of dynamic partner-partner interaction (i.e., interactive adaptation hypothesis). The final model allowed for dynamic changes in individual motor plans (and therefore, movement vigor) based on partner-partner interactions and observations. After detailed observations of interaction torque and movement duration (or, vigor), the authors concluded that the interactive adaptation model provided the best explanation of human-human interaction during self-paced dyadic movements.

Strengths:

The experimental setup (simultaneous wrist extension-flexion movements) has been thoroughly vetted. The task was designed particularly well, with adequate block pseudo-randomization to ensure general validity of the results. The analyses of torque interaction, movement kinematics, and vigor are sound, as are the statistical measures used to assess significance. The authors structured the work via a helpful comparison of several candidate models of human-human interaction dynamics, and how well said models explained variance in the vigor of solo and combined movements.

The authors adequately addressed several concerns that I raised in my initial review of the work, including clarity regarding analyses of movement vigor and inclusion of additional analyses of reaction time. The results are supported by both parametric and non-parametric statistical methods.

The research question is timely and extends current neuroscientific understanding of sensorimotor control, particularly in social contexts. This work answers several new, important questions about control of vigor during volitional movements, and in doing so it motivates future research into the topic.

Weaknesses:

My chief concern about the study is the relatively low number of dyad data points (n=10). The authors recruited 20 participants, but the primary conclusions are based on dyad-specific interactions (i.e., analyses of "fast" vs "slow" participants in each pair). However, it is important to note that most of the effects upon which the conclusions rest are associated with relatively large effect sizes.

---

## [Referee Report · Reviewer #2 (Public review)]

Summary:

This study examines how individual movement vigor is integrated into a shared, dyadic vigor when two individuals are physically coupled. Participants performed wrist-reaching movements toward targets at different distances while mechanically linked via a virtual elastic band, and dyads were formed by pairing participants with different baseline vigor profiles. Under interaction conditions, movements converged to coordinated patterns that could not be explained by simple averaging, indicating that each dyad behaved as a single functional unit. Notably, under coupling, movement durations for both partners were shorter than in the solo condition, arguing against the view that each individual simply executed an independent movement plan. Furthermore, dyadic vigor was primarily predicted by the slower partner's vigor rather than by the faster partner's, suggesting that neither a leader-follower strategy nor a weighted averaging account fully explains the observed behavior. The authors propose a computational model in which both partners adapt to the emerging interaction dynamics ("interactive adaptation strategy"), providing a coherent explanation of the behavioral observations.

Strengths:

The study is carefully designed and addresses an important question about how individual movement vigor is integrated during joint action. The experimental paradigm allows systematic manipulation of interaction strength and partner asymmetry. The behavioral results show clear and robust patterns, particularly the shortening of movement durations under elastic coupling (KL and KH condition) and the asymmetrical contribution of the slower partner's vigor to dyadic vigor. The computational model captures the main behavioral patterns well and provides a principled framework for interpreting dyadic vigor not as a simple combination of two independent motor plans, but as an emergent property arising from mutual adaptation. Conceptually, the study is notable in extending the notion of vigor from an individual attribute to a dyad-level construct, opening a new perspective on coordinated movement and motor decision-making.

Weaknesses:

The revised manuscript now clearly explains why the proposed computational model successfully accounts for the observed dyadic behavior. In particular, the mechanisms by which uncertainty associated with the slower partner and time-related costs of the faster partner jointly shape dyadic vigor are now clear. I have no further comments to add.

---

## [Referee Report · Reviewer #3 (Public review)]

Summary:

This study provides novel insights into how individuals regulate the speed of their movements both alone and in pairs, highlighting consistent differences in movement vigor across people and showing that these differences can adapt in dyadic contexts. The findings are significant because they reveal stable individual patterns of action that are flexible when interacting with others, and they suggest that multiple factors, beyond reward sensitivity, may contribute to these idiosyncrasies. The evidence is generally strong, supported by careful behavioral measurements and appropriate modeling.

The authors have addressed all of my previous comments. I appreciate the clarification of abbreviations, terminology, and key concepts, the expansion of the discussion, and the adjustments to some of the statistical analyses in response to both my earlier comments and those of Reviewer 1.

---

## [Author Response]

The following is the authors’ response to the original reviews.

We thank the reviewers for their constructive and precise comments, which have helped us improve the consistency and clarity of our manuscript. Below, we provide a point-by-point response to each comment. In summary, the main changes introduced in the revised version are as follows:

(1) We replaced all the statistical analyses to their non-parametric equivalents to ensure compliance with test assumptions and consistency of the results;

(2) We compare the participants’ reaction times before and during connected practice, revealing a significant reduction in reaction times of both partners when connected;

(3) We added, in the supplementary materials, a table reporting the vigor scores of each participant in each experimental condition, facilitating the assessment of individual and dyadic behaviors;

(4) We have reviewed and refined the terminology throughout the manuscript and reduced the number of abbreviations to improve clarity.

**Public Reviews:**

**Reviewer #1 (Public review):**
Summary:The authors present a novel investigation of the movement vigor of individuals completing a synchronous extension-flexion task. Participants were placed into groups of two (so-called "dyads") and asked to complete shared movements (connected via a virtual loaded spring) to targets placed at varying amplitudes. The authors attempted to quantify what, if any, adjustments in movement vigor individual participants made during the dyadic movements, given the combined or co-dependent nature of the task. This is a novel, timely question of interest within the broader field of human sensorimotor control.Participants from each dyad were labeled as "slow" (low vigor) or "fast" (high vigor), and their respective contributions to the combined movement metrics were assessed. The authors presented four candidate models for dyad interactions: (a) independent motor plans (i.e., co-activity hypothesis), (b) individual-led motor plans (i.e., leader-follower hypothesis), (c) generalization to a weighted average motor plan (i.e., weighted adaptation hypothesis), and (d) an uncertainty-based model of dynamic partner-partner interaction (i.e., interactive adaptation hypothesis). The final model allowed for dynamic changes in individual motor plans (and therefore, movement vigor) based on partner-partner interactions and observations. After detailed observations of interaction torque and movement duration (or vigor), the authors concluded that the interactive adaptation model provided the best explanation of human-human interaction during self-paced dyadic movements.Strengths:The experimental setup (simultaneous wrist extension-flexion movements) has been thoroughly vetted. The task was designed particularly well, with adequate block pseudo-randomization to ensure general validity of the results. The analyses of torque interaction, movement kinematics, and vigor are sound, as are the statistical measures used to assess significance. The authors structured the work via a helpful comparison of several candidate models of human-human interaction dynamics, and how well said models explained variance in the vigor of solo and combined movements. The research question is timely and extends current neuroscientific understanding of sensorimotor control, particularly in social contexts.

We thank the reviewer for their in-depth analysis and constructive assessment of our manuscript.

Weaknesses:(1) My chief concern about the study as it currently stands is the relatively low number of data points (n=10). The authors recruited 20 participants, but the primary conclusions are based on dyad-specific interactions (i.e., analyses of "fast" vs "slow" participants in each pair). Some of these analyses would benefit greatly, in terms of power, from the addition of more data points.

We understand and appreciate the reviewer’s concern regarding the effective sample size at the dyad level (n=10). While our primary analyses focus on dyad-specific interactions, we note that the reported effects are consistent across multiple dynamic conditions and are associated with large effect sizes. To provide a conservative assessment the Cohen’s *D* values reported correspond to the smallest effect size observed across the relevant statistical tests, thereby limiting the risk of false positives or overinterpretation. In addition, to ensure robustness given the sample size and distribution properties of the data, we have replaced all parametric tests with their non-parametric counterparts, as some analyses violated ANOVA assumptions. Friedman and Kruskal-Wallis tests are now used for paired and unpaired main effects respectively, and Wilcoxon and Mann-Whitney tests for paired and unpaired post-hoc comparisons respectively. Note that these changes did not alter the conclusions of the study.

(a) The distribution of delta-vigor (Fast group vs Slow group) is highly skewed (see Figures 3D, S6D), with over half of the dyads exhibiting delta-vigor less than 0.2 (i.e., less than 20% of unit vigor). Given the relatively low number of dyads, it would be helpful for the authors to provide explicit listings of VigorFast, VigorSlow, and VigorCombined for each of the 10 separate dyads or pairings.

We agree with this comment. However, we note that the distribution of vigor scores within a population is typically centered around 1, with large deviations observed only for the fastest and slowest participants [1]. As a result, the distri bution of ∆-vigor is inherently skewed. Correcting for this skewness would (i) require pairing participants based on their vigor, which is logistically difficult, and (ii) lead to an atypical sampling of dyads, with an over representation of pairs exhibiting very large vigor differences. The distributions of vigor scores for the fast and slow groups before and after the interaction are reported in Supplementary Fig. S21. In addition, as suggested by the reviewer, we have now included Table S.1 in the supplementary materials, listing the values VigorFast, VigorSlow, and VigorCombined for each of the 10 dyads. This table provides a complete view of the evolution of participant’s vigor throughout the experiment.

(b) The authors concluded that the interactive adaptation hypothesis provided the best summary of the combined movement dynamics in the study. If this is indeed the case, then the relative degree of difference in vigor between the fast and slow participants in a dyad should matter. How well did the interactive adaptation model explain variance in the dyads with relatively low delta-vigor (e.g., less than 0.2) vs relatively high delta-vigor?

We initially expected the magnitude of difference in individual vigor within a dyad to play a significant role. However, our analysis did not reveal any systematic effect of ∆-vigor on either the interaction force or the resulting dyadic vigor, as shown by the LMM analysis. Importantly, the interactive adaptation hypothesis does per se imply that the magnitude of vigor differences between the two partners should matter, only that their respective roles in selecting the adapted behavior is different. Although the model includes several free parameters, we did not attempt to fit it to individual dyads as would in principle be possible. Instead, we performed a sensitivity analysis to assess how variations in the difference in vigor between the partners influence model predictions. For this purpose, we simulated increasing values of *µ* and variations in the fast partner’s cost of time. In addition, we demonstrated that uncertainty in the estimated behavior of the slow partner, which is a priori specific to each individual, has a substantial impact on the optimal movement duration of the dyad. Overall, this analysis shows that the model captures the full range of qualitative trends observed in the experimental data. When applied to predict the behavior of the average dyad, the resulting movement time prediction error remain small, as detailed in the Results section.

(2) The authors shared the results of one analysis of reaction time, showing that the reaction times of the slow partners and the fast partners did not differ during the initial passive block. Did the authors observe any changes in RT of either the slow or fast partner during the combined (primary task) blocks (KL, KH, etc.)? If the pairs of participants did indeed employ a form of interactive adaptation, then it is certainly plausible that this interaction would manifest in the initial movement planning phase (i.e., RT) in addition to the vigor and smoothness of the movements themselves.

We thank the reviewer for this interesting question, that prompted us to extend our analysis of reaction times to the connected conditions. This additional analysis revealed a significant main effect of the condition on the reaction time for both the fast and slow groups (in both cases: *W*_2_ > 0.39, *p* < 0.02). Post-hoc comparisons showed a significant reduction in reaction time between the initial null-field block (NF1) and the KH condition for the slow group (*p* = 0.03, *D* = 1.46), and a similar trend for the fast group (*p* = 0.06, *D* = 1.03). However, the reaction times remained comparable between the two groups, with no significant difference between them. We have incorporated these observations in the Results section (p.4, l.100–109) and expanded the Discussion (p.11, l.341–348) to address their implications for interactive adaptation in human-human and human-robot physical interactions.

**Reviewer #2 (Public review):**
Summary:This study examines how individual movement vigor is integrated into a shared, dyadic vigor when two individuals are physically coupled. Participants performed wrist-reaching movements toward targets at different distances while mechanically linked via a virtual elastic band, and dyads were formed by pairing participants with different baseline vigor profiles. Under interaction conditions, movements converged to coordinated patterns that could not be explained by simple averaging, indicating that each dyad behaved as a single functional unit. Notably, under coupling, movement durations for both partners were shorter than in the solo condition, arguing against the view that each individual simply executed an independent movement plan. Furthermore, dyadic vigor was primarily predicted by the slower partner’s vigor rather than by the faster partner’s, suggesting that neither a leader-follower strategy nor a weighted averaging account fully explains the observed behavior. The authors propose a computational model in which both partners adapt to the emerging interaction dynamics ("interactive adaptation strategy"), providing a coherent explanation of the behavioral observations.Strengths:The study is carefully designed and addresses an important question about how individual movement vigor is integrated during joint action. The experimental paradigm allows systematic manipulation of interaction strength and partner asymmetry. The behavioral results show clear and robust patterns, particularly the shortening of movement durations under elastic coupling (KL and KH conditions) and the asymmetrical contribution of the slower partner’s vigor to dyadic vigor. The computational model captures the main behavioral patterns well and provides a principled framework for interpreting dyadic vigor not as a simple combination of two independent motor plans, but as an emergent property arising from mutual adaptation. Conceptually, the study is notable in extending the notion of vigor from an individual attribute to a dyad-level construct, opening a new perspective on coordinated movement and motor decision-making.

We thank the reviewer for their thorough analysis of our manuscript and their constructive feedback.

Weaknesses:(1) A key conceptual issue concerns the apparent asymmetry between partners in the computational framework. While dyadic vigor is empirically better predicted by the slower partner’s vigor, the model formulation appears to emphasize the faster partner’s time-related cost and interaction forces. Although the cost function includes an uncertaintyrelated component associated with the slower partner, it remains unclear from the current formulation and description how dyadic vigor is formally derived from the slower partner’s control policy within the same modeling framework. This raises an important question regarding whether the model offers a symmetric account of dyadic vigor formation for both partners or whether it is effectively anchored to the faster partner’s control architecture.

We have modified our phrasing to clarify the principles according to which the computational framework was designed (p.7, l.226–231 and p.9, l.260–264). As stated in the Results section, the model is indeed asymmetric by design, which corresponds to the different roles of the fast and slow partner exhibited in the data. In that context, the uncertain term associated with the slow partners should be understood as an overarching constraint that conditions the strategy of the dyad, while the fast partner cost of time acts as a contributor to the expected dyad strategy. Conceptually and numerically as reported in the sensitivity analysis, this asymmetry corresponds to the role of the slow partners in setting the vigor ranking among the dyads and the role of the fast partner in setting the average dyadic behavior.

(2) A second conceptual issue concerns the interpretation of the term "motor plan." It remains unclear whether this term refers primarily to movement-related characteristics such as speed or duration, or more broadly to the underlying optimization structure that governs these variables. This distinction is theoretically important, as it determines whether the reported interaction effects should be understood as adjustments in movement characteristics or as changes in the structure of the control policy itself.

We agree with the reviewer that this terminology required clarification. In this paper, the term “motor plan” refers to the time series of control inputs planned by the CNS, rather than solely to kinematic descriptors such as speed or duration. These planned control signals are a direct consequence of the underlying optimization structure and cost functions that govern trajectory generation. We have clarified this definition in the Introduction (p.1, l.23–24).

**Reviewer #3 (Public review):**
Strengths:This study provides novel insights into how individuals regulate the speed of their movements both alone and in pairs, highlighting consistent differences in movement vigor across people and showing that these differences can adapt in dyadic contexts. The findings are significant because they reveal stable individual patterns of action that are flexible when interacting with others, and they suggest that multiple factors, beyond reward sensitivity, may contribute to these idiosyncrasies. The evidence is generally strong, supported by careful behavioral measurements and appropriate modeling, though clarifying some statistical choices and including additional measures of accuracy and smoothness would further strengthen the support for the conclusions.

Thank you for this analysis and the insightful feedback.

Major Comments:(1) Given the idiosyncrasies in individual vigor, would linear mixed models (LMMs) be more appropriate than ANOVAs in some analyses (e.g., in the section "Solo session"), as they can account for random intercepts and slopes on vigor measures? Some figures (e.g., Figure 2.B and 3.E) indeed seem to show that some aspects of behaviour may present variability in slopes and intercepts across participants. In fact, I now realize that LMMs are used in the "Emergence of dyadic vigor from the partners’ individual vigor" section, so could the authors clarify why different statistical approaches were applied depending on the sections?

We thank the reviewer for this thoughtful comment. We deliberately used different statistical approaches throughout the paper in order to address different types of questions. Note that the statistical tests were converted to their nonparametric equivalent for consistency (see answer to Reviewer 1).

- Friedman tests were used in a limited number of cases to assess population- or group-level effects, such as differences in movement time, smoothness, or accuracy across the solo, connected, and after-effects conditions. Such tests provide a straightforward framework for these descriptive, condition-level comparisons.

- The stability of individual and dyadic vigor scores across conditions was assessed using Pearson correlations across all condition pairs, which we consider the most direct and interpretable approach for evaluating consistency across sessions.

- LMMs were employed to examine how dyadic vigor relates to the partners’ individual vigor measured in the solo conditions, which revealed the critical contribution of the slow partner.

Rather than applying a single statistical framework throughout, we selected the method best suited to each question. While LMMs are well suited for modeling participant-specific variability when linking individual and dyadic measures, their systematic use in all analyses would be less intuitive and would not directly address several of the population-level comparisons central to this study.

(2) If I understand correctly, the introduction suggests that idiosyncrasies in movement vigor may be driven by interindividual differences in reward sensitivity. However, the current task does not involve any explicit rewards, yet the authors still observe idiosyncrasies in vigor, which is interesting. Could this indicate that other factors contribute to these consistent individual differences? For example, could sensitivity to temporal costs or physical effort explain the slow versus fast subgrouping? Specifically, might individuals more sensitive to temporal costs move faster to minimize opportunity costs, and might those less sensitive to effort costs also move faster? Along the same lines, could the two subgroups (slow vs. fast) be characterized in terms of underlying computational "phenotypes," such as their sensitivities to time and effort? If this is not feasible with the current dataset, it would still be valuable to discuss whether these factors could plausibly account for the observed patterns, based on existing literature.

We thank the reviewer for this interesting question. We first note that the notion of reward in motor control is quite broad. Although our task did not include explicit external (e.g. monetary) rewards, we assumed that participants attribute an implicit value to completing the task in accordance with the experimenter’s instructions. This assumption has been shown to be appropriate for characterising baseline behavior in previous studies [2–5].

As discussed in the Introduction, vigor is generally understood to emerge from a tradeoff between effort, accuracy, and time. The reviewer is correct in noting that inter-individual differences in vigor may reflect differences in reward sensitivity or in its discounting [3,6], given that time and reward are intrinsically coupled. Differences in vigor may also arise from inter-individual variability in sensitivity to effort or perceived task difficulty. Because these factors are intertwined—for example, increasing accuracy through co-contraction typically incurs greater effort [7]—it is challenging to disentangle their respective contributions based solely on behavioral data.

In the present study, our inverse optimal control procedure to identify the cost of time (and thus predict individuals’ vigor) relies on a predefined effort-accuracy tradeoff under fixed final time across multiple movement amplitudes [8]. As a result, the model does not allow us to independently estimate individual sensitivities to effort, accuracy, and time. Such characterization of computational "phenotypes" would likely require experimental paradigms in which each of these factors is systematically manipulated while the others are held constant, which is beyond the scope of the current dataset. In practice, the main value of behavioral modeling lies in revealing the relative weighting of these criteria by the CNS during motor planning [5]. We have expanded the Discussion to clarify these limitations and considerations (see Discussion p.12, l.396–401 & l.407–412).

Finally, we chose not to emphasize these broader issues in the present manuscript because (i) they are peripheral to our primary research question on how individual vigor influences human-human interaction, and (ii) although we do not yet have definitive and consensual answers, they have been addressed in multiple studies reviewed elsewhere [9,10].

(3) The observation that dyads did not lose accuracy or smoothness despite changes in vigor is interesting and suggests a shift in the speed-accuracy tradeoff. Could the authors include accuracy and smoothness measures in the main figures rather than only in supplementary materials? I think it would make the manuscript more complete.

We also find that the preservation of accuracy and smoothness despite changes in vigor is an interesting result, and we therefore chose to report these measures in the Supplementary Materials. However, we believe it is preferable not to include them in the main figures for the following reasons:

- We avoid framing our results in terms of a speed-accuracy trade-off, as Fitts’ work was initially designed to study fast movements [11], whereas our work focuses on self-paced movements. As outlined in the Introduction, vigor is more appropriately interpreted as reflecting a tradeoff between effort (related to movement speed), accuracy, and time. From this perspective, the reported changes of vigor already capture a shift in the underlying trade-off selected by the CNS, using a framework better suited to our experimental paradigm.

- The manuscript is technically dense and reports multiple analyses that are essential to establish (i) the existence and definition of dyadic vigor, and (ii) how it emerges from interaction between partners. Although the observed preservation of accuracy and improvements in smoothness are informative, they are not central to these two primary questions and would risk diverting attention from the core contributions of the paper. In addition, accuracy is not a feature predicted by our deterministic modeling and extensions would be needed to capture these aspect. Here we only attempted to replicate average behaviors.

(4) It is a bit unclear to me whether the variance assumptions for ANOVAs were checked, for instance, in Figure 3H.

We thank the reviewer for this comment, which prompted us to verify the assumptions underlying our ANOVAs. We found that a few distributions in the original analysis, as well as in some of the new tests, did not meet these assumptions. To ensure consistency, all statistical analyses have now been replaced with non-parametric tests: Friedman and Kruskal-Wallis tests for paired and unpaired main effects, Wilcoxon and Mann-Whitney tests for paired and unpaired post-hocs. The updated results do not change any of the conclusions. the only minor change is accuracy, that appeared slightly improved in a restricted number of connected conditions, and now appears mostly non-impacted.

**Recommendations for the authors:**

**Reviewer #1 (Recommendations for the authors):**
Minor points:(1) Lines 146-147. The authors state, "Whereas the fast partners maintained a similar duration". Figures S6H,I suggest that fast partners made slower movements during the paired task relative to the solo task, not movements with a similar duration.

We agree that Fig. S.6H,I suggest slightly slower movements for the fast partners, though not significant. We have modified the sentence to be less assertive than in the previous version (see p.6, l.155).

(2) In the Discussion (Lines 318-319), the authors state that their findings confirm and extend the "benefits of dyadic control in collaborative actions". What benefits are they referring to here, relative to individual control? It would be helpful if the authors would elaborate on this claim.

We have modified this sentence to clarify that the benefits of dyadic control refer to previously reported advantages over individual control, namely reduced movement time Reed and Peshkin (2008) [12] and improved tracking accuracy [13,14] (see p.11, l.336–337).

(3) On Lines 87-89, the authors reference a decomposition of variance of vigor scores across the NF1, VL, and VH conditions; however, I did not see an explanation of how this decomposition was performed. The method used to estimate variance explained by inter-individual vs intra-individual differences in vigor should be outlined for the reader.

Thank you for pointing out this missing information. We now explain in the statistical analysis section (see p.14, l.504–507), that the percentage of inter-individual variability in vigor is estimated using sum-square values as an estimation of inter- and intra-individual variability.\begin{document}$$\displaystyle \operatorname{Var}_{\text {inter }}=100 \frac{\mathrm{SS}_{\text {inter }}}{\mathrm{SS}_{\text {inter }}+\mathrm{SS}_{\text {intra }}}$$\end{document}

(4) How was the absolute interaction torque for a paired movement calculated? Was it an integral of the temporal profile of torque for some portion of the combined movement? The method for calculating the absolute interaction torque needs to be specified.

We have now clarified in the Methods (see p.14, l.490–491) that the reported average interaction effort was computed as the absolute value of the interaction torque as a function of time averaged over the entire movement.

(5) Lines 123-124: "... interaction torque showed no significant correlation with differences in individual vigor within dyads." This statement should be supported by appropriate statistical measures.

This result is now supported by reporting the corresponding Pearson correlation analyses. No significant correlations were found between interaction torque and differences in individual vigor within dyads (KL conditions: |*r*| < 0.43, *p*> 0.22; KH conditions: |*r*| < 0.18, *p* > 0.61, see p.5, l.132–133).

(6) For the analysis, presented in Figure 3C, and specified on lines 116-123, the text mentions the main effects of both condition and target. There doesn’t appear to be much of an effect of the target for the KH data. Should these results not be reported as an interaction effect between the two factors instead?

We agree with the reviewer and have corrected our presentation of these results (see p.4, l.126–128). Consistent with the reviewer’s observation, no significant effect of the target is found in the KH condition.

(7) Figures 3E and S6B. What is the purpose of including the averaged data for each pair in addition to both individuals’ data from each pair? It would be useful to distinguish the individual data from the average data for each pair. Frankly, the number of data points shown on this sub-figure is excessive.

There may have been a misunderstanding. Because the partners of a dyad are connected by a virtual elastic band (rather than a rigid bar), they do not execute identical movements. Therefore Figs. 3E,S6B display the movement time of all individual participants, together with the corresponding 20 individual regression lines, like in Fig. 2B. The solid black line represents the average across all individuals, and the averaged behaviors of dyads are not included. We have clarified this point by revising the caption of Fig. 3E (see p.5).

Noted mis-spellings:Figure S.3A caption: "trials towards this target."Page 10 Line 313: "Importantly, these findings show ...".

These mis-spellings have been corrected at supplementary p.2 and main text p.11, l.331. Thank you!

**Reviewer #2 (Recommendations for the authors):**
(1) To illustrate the contribution of the three components used to calibrate the overall cost function, it would be informative to include simulation analyses in which each component is selectively removed (i.e., ablation analyses).

We did not perform ablation analyses, as selectively removing components of the model can lead to instability or ill-suited control inputs, making the resulting simulations difficult to interpret. Instead, we conducted a sensitivity analysis of the key parameters shaping the overall cost function, including the estimated mean and deviation of the slow partner’s movement duration, the weight associated with uncertain torque minimization (Figs. S.18,S.19), and the fast partner’s cost of time (Fig. S20). This analysis reveals the predominant roles of the estimated slow partner movement patterns in determining the model predictions, in agreement with our experimental observations.

(2) Although the authors refer to the motor-off condition as "passive," participants actively generated the movements in the absence of external forces. Thus, this condition corresponds to active, unassisted movement. A different term may therefore reduce potential confusion for readers.

We agree that term “passive” was not well-chosen given the context of the paper, thus we have instead replaced this denomination as “null-field” condition. Consequently, the P1 and P2 blocks are now referred to as NF1 and NF2.

(3) Please clarify the instructions given to participants. Were they informed in advance that their movements would physically interact with those of their partner?

Thank you for pointing out this missing clarification. We have now specified in the Methods (p.14, l.465–469) that participants were not informed prior to any condition that they would interact with a human partner; they were only told that the robot would provide assistance. When debriefed at the end of the experiment, only one out of the 20 participants reported having realized that they were connected to another human. Most participants believed they were interacting either with a version of themselves or with a robot with some randomness.

(4) Line 475. Should "Fig. 2D" be "Fig. 2B"?

Thank you for catching this error. The reference has been corrected to Fig. 2B (see p.15, l.522).

**Reviewer #3 (Recommendations for the authors):**
(1) The analysis of reaction times shows no difference between groups in the passive block, which challenges the assumption that movement vigor covaries with decision speed or action initiation speed. It may be worth discussing this in the context of recent literature.

We agree that the initial analysis and discussion of reaction times were too superficial. In the revised manuscript, we now report that dyadic interaction leads to significantly shorter reaction times (p.4, l.100–109), concomitantly with improved movement velocity. We have also expanded the Discussion, on the relationship between decision and action speeds/durations (p.11, l.340–348).

(2) Many abbreviations are unusual for a non-expert. I would recommend using the full terms instead. At least initially, I found it difficult to follow the results because the abbreviations were not immediately clear (at least to me).

We agree that the paper had to many abbreviations. Therefore, we have removed the abbreviated names of the models and, when possible without impacting the readability, used the full names of the conditions.

(3) Relatedly, the notation in Figure 1 may be confusing. The labels "S" and "F" (slow and fast) correspond to different concepts than "F" and "L" (follower and leader), so the same participant could be labeled "F" as fast but not "F" as a leader.

Thank you for pointing out this potential source of confusion. We have therefore modified Fig. 1A (p.2) to avoid any potential confusion by using the full model names rather than abbreviations. In the remainder of the manuscript, "S" and "F" exclusively denote the slower and faster partners within a dyad, and we do not use abbreviations for "leader" or "follower" in the text.

(4) In figures like 2.C and 3.I, keeping the same scales on the x and y axes and adding a diagonal reference line would make it easier to see shifts across conditions.

As explained in the Methods, vigor scores in the low- and high-viscosity conditions were computed using the average movement durations from the NF1 condition as a reference. Consequently, because movements are slower in these conditions, the corresponding vigor values are lower than those in NF1. For this reason, using identical scales on the x- and y-axes and adding a 45◦ reference line could mislead the reader in thinking that the vigor scores are expected to be identical and reduce the readability of the figure.

(5) Multiple hypotheses about dyadic regulation of vigor are nicely explained; it could help to indicate if any of these were a priori favored based on prior literature.

Previous literature provides mixed evidence regarding how vigor might be regulated in dyadic interaction. For instance, Takagi et al. (2016) [15] reported that mechanically connected partners may rely on independent motor plans, which corresponds to the co-activity hypothesis considered here. However, in that study, movement duration was prescribed. We therefore expected that removing this constraint on movement duration could allow coordination strategies to emerge, particularly in view of findings on haptic communication during tracking of random targets while connected via an elastic band [13,14].

At the same time, a large body of work on human–human and human–robot interaction has interpreted coordination through a leader–follower framework. In our context, vigor is understood as the outcome of a tradeoff between effort and elapsed time, with time being associated with a decaying reward. Based on this framework, we hypothesized a priori that a leader–follower scheme would emerge, in which the fast partner—being more sensitive to time costs and/or less sensitive to effort—would tend to drive the interaction, even at the expense of increased effort. For these reasons, the leader–follower hypothesis was formulated as the expected outcome throughout the manuscript.

(6) In the introduction, statements such as "relative vigor of an individual is remarkably stable" appear true only in the solo condition. The same is true in the discussion where it is said that vigor is a stable trait. The whole study show that an individual can shift his/her vigor to the same vigor of another individual, so it doesn’t appear stable to me in such conditions but adaptable.

Let us first clarify that when we describe vigor as “remarkably stable”, we do not imply that individuals do not adjust their movement timing in response to changes in external dynamics. For example, movement durations increase in visco-resistive conditions even during solo performance; nevertheless, individuals who move faster in the absence of resistance will remain faster relative to others when resistance is introduced. In this sense, stability refers to the preservation of relative rankings across conditions, rather than invariance of absolute movement timing. Because interaction with another individual constitutes a substantial change in task dynamics, an effect on individual pace is therefore expected.

Told that (and as pointed to by the reviewer) (i) dyadic interactions lead to the emergence of a dyadic vigor characterized by average movement durations close to those of the fast partners, while the ranking across dyads is largely imposed by the slow partners; and (ii) these adaptations persist after the interaction phase. Importantly, the observed vigor adaptations appear to last longer in our physical interaction task than in previous attempts to manipulate vigor using visual feedback [16]. To account for this adaptability of vigor, we have (i) clarified claims in the Introduction regarding the stability of vigor (see p.1, l.18–20), and (ii) expanded the Discussion to more explicitly address vigor adaptability and the possible resulting consequences for the concept of vigor (see p.12, l.407–412).

References

(1) O. Labaune, T. Deroche, C. Teulier, and B. Berret, “Vigor of reaching, walking, and gazing movements: on the consistency of interindividual differences,” Journal of Neurophysiology, vol. 123, pp. 234–242, jan 2020.

(2) L. Rigoux and E. Guigon, “A model of reward-and effort-based optimal decision making and motor control,” PLoS Computational Biology, vol. 8, pp. 1–13, Jan. 2012.

(3) R. Shadmehr, J. J. O. de Xivry, M. Xu-Wilson, and T.-Y. Shih, “Temporal discounting of reward and the cost of time in motor control,” Journal of Neuroscience, vol. 30, pp. 10507–10516, aug 2010.

(4) B. Berret and G. Baud-Bovy, “Evidence for a cost of time in the invigoration of isometric reaching movements,” Journal of Neurophysiology, vol. 127, pp. 689–701, feb 2022.

(5) D. Verdel, O. Bruneau, G. Sahm, N. Vignais, and B. Berret, “The value of time in the invigoration of human movements when interacting with a robotic exoskeleton,” Science Advances, vol. 9, sep 2023.

(6) K. Jimura, J. Myerson, J. Hilgard, T. S. Braver, and L. Green, “Are people really more patient than other animals? evidence from human discounting of real liquid rewards,” Psychonomic Bulletin & Review, vol. 16, pp. 1071–1075, dec 2009.

(7) P. L. Gribble, L. I. Mullin, N. Cothros, and A. Mattar, “Role of cocontraction in arm movement accuracy,” Journal of Neurophysiology, vol. 89, pp. 2396–2405, may 2003.

(8) B. Berret and F. Jean, “Why Don’t We Move Slower? The Value of Time in the Neural Control of Action,” Journal of Neuroscience, vol. 36, pp. 1056–1070, Jan. 2016.

(9) R. Shadmehr and A. A. Ahmed, Vigor : neuroeconomics of movement control. The MIT Press, 2020.

(10) D. Thura, A. M. Haith, G. Derosiere, and J. Duque, “The integrated control of decision and movement vigor,” Trends in Cognitive Sciences, vol. 29, pp. 1146–1157, Dec. 2025.

(11) P. M. Fitts, “The information capacity of the human motor system in controlling the amplitude of movement,” Journal of Experimental Psychology, vol. 47, pp. 381–391, June 1954.

(12) K. B. Reed and M. A. Peshkin, “Physical collaboration of human-human and human-robot teams,” IEEE Transactions on Haptics, vol. 1, pp. 108–120, July 2008.

(13) G. Gowrishankar, A. Takagi, R. Osu, T. Yoshioka, M. Kawato, and E. Burdet, “Two is better than one: physical interactions improve motor performance in humans,” Scientific Reports, vol. 4, Jan. 2014.

(14) A. Takagi, G. Ganesh, T. Yoshioka, M. Kawato, and E. Burdet, “Physically interacting individuals estimate the partner’s goal to enhance their movements,” Nature Human Behaviour, vol. 1, pp. 1–6, Mar. 2017.

(15) A. Takagi, N. Beckers, and E. Burdet, “Motion plan changes predictably in dyadic reaching,” PLOS ONE, vol. 11, p. e0167314, Dec. 2016.

(16) P. Mazzoni, B. Shabbott, and J. C. Cortes, “Motor control abnormalities in Parkinson’s disease,” Cold Spring Harbor Perspectives in Medicine, vol. 2, pp. a009282–a009282, Mar. 2012.